# Ultrastructural Study and Immunohistochemical Characteristics of Mesencephalic Tegmentum in Juvenile Chum Salmon (*Oncorhynchus keta*) Brain After Acute Traumatic Injury

**DOI:** 10.3390/ijms26020644

**Published:** 2025-01-14

**Authors:** Evgeniya V. Pushchina, Evgeniya A. Pimenova, Ilya A. Kapustyanov, Mariya E. Bykova

**Affiliations:** A.V. Zhirmunsky National Scientific Center of Marine Biology, Far Eastern Branch, Russian Academy of Sciences, 690041 Vladivostok, Russia; eapimenova@yandex.ru (E.A.P.); ilyaak9772@gmail.com (I.A.K.); stykanyova@mail.ru (M.E.B.)

**Keywords:** tegmentum, *Oncorhynchus keta*, transmission electron microscope, dorsal tegmental nucleus, nucleus of fasciculus longitudinalis medialis, nucleus of oculomotor nerve, brain lipid-binding protein, aromatase B

## Abstract

The ultrastructural organization of the nuclei of the tegmental region in juvenile chum salmon (*Oncorhynchus keta*) was examined using transmission electron microscopy (TEM). The dorsal tegmental nuclei (DTN), the nucleus of *fasciculus longitudinalis medialis* (NFLM), and the nucleus of the oculomotor nerve (NIII) were studied. The ultrastructural examination provided detailed ultrastructural characteristics of neurons forming the tegmental nuclei and showed neuro–glial relationships in them. Neurons of three size types with a high metabolic rate, characterized by the presence of numerous mitochondria, polyribosomes, Golgi apparatus, and cytoplasmic inclusions (vacuoles, lipid droplets, and dense bodies), were distinguished. It was found that large interneurons of the NFLM formed contacts with protoplasmic astrocytes. Excitatory synaptic structures were identified in the tegmentum and their detailed characteristic are provided for the first time. Microglia-like cells were found in the NIII. The ultrastructural characteristics of neurogenic zones of the tegmentum of juvenile chum salmon were also determined for the first time. In the neurogenic zones of the tegmentum, adult-type neural stem progenitor cells (aNSPCs) corresponding to cells of types III and IVa Danio rerio. In the neurogenic zones of the tegmentum, neuroepithelial-like cells (NECs) corresponding to cells previously described from the zebrafish cerebellum were found and characterized. In the tegmentum of juvenile chum salmon, patterns of paracrine neurosecretion were observed and their ultrastructural characteristics were recorded. Patterns of apoptosis in large neurons of the tegmentum were examined by TEM. Using immunohistochemical (IHC) labeling of the brain lipid-binding protein (BLBP) and aromatase B (AroB), patterns of their expression in the tegmentum of intact animals and in the post-traumatic period after acute injury to the *medulla oblongata* were characterized. The response to brainstem injury in chum salmon was found to activate multiple signaling pathways, which significantly increases the BLBP and AroB expression in various regions of the tegmentum and *valvula cerebelli*. However, post-traumatic patterns of BLBP and AroB localizations are not the same. In addition to a general increase in BLBP expression in the tegmental parenchyma, BLBP overexpression was observed in the rostro-lateral tegmental neurogenic zone (RLTNZ), while AroB expression in the RLTNZ was completely absent. Another difference was the peripheral overexpression of AroB and the formation of dense reactive clusters in the ventro-medial zone of the tegmentum. Thus, in the post-traumatic period, various pathways were activated whose components were putative candidates for inducers of the “astrocyte-like” response in the juvenile chum salmon brain that are similar to those present in the mammalian brain. In this case, BLBP acted as a factor enhancing the differentiation of both radial glia and neurons. Estradiol from AroB+ astrocytes exerted paracrine neuroprotective effects through the potential inhibition of inflammatory processes. These results indicate a new role for neuronal aromatization as a mechanism preventing the development of neuroinflammation. Moreover, our findings support the hypothesis that BLBP is a factor enhancing neuronal and glial differentiation in the post-traumatic period in the chum salmon brain.

## 1. Introduction

Tegmental area of the fish brain has been insufficiently studied compared to other areas: the telencephalon [1,2,3], cerebellum [4,5,6], and *tectum opticum* [7,8,9]. Of particular interest are the neurogenic areas of the fish brain tegmentum, since regions containing adult-type neural stem progenitor cells (aNSPCs) have been found there [9,10]. The midbrain tegmentum and isthmus in Pacific salmon contain a substantial number of cellular populations [11,12], but not all of them have been identified as having possible homologs and as those that have undergone an evolutionary transformation. One of the explanations for this is the reticular origin of most of these structures, since some tegmental nuclei are part of the reticular formation, while others, highly developed in the brain of higher vertebrates, are probably its derivatives.

In the fish brain, aNSPCs are located in distinct neurogenic niches that vary in their location, cellular composition, and proliferative behavior [13]. The heterogeneity of the aNSPC population is assumed to reflect differential capacities for neurogenesis, plasticity, and repair between different neurogenic zones. Following investigations of neurogenesis in humans, studies have focused primarily on the behavior and biological significance of aNSPCs in rodent models [14]. However, unlike rodents, which exhibit neurogenesis in only two limited neurogenic niches throughout life, constitutive neurogenesis in fish has been described from multiple stem cell niches that provide new neurons to each major brain region [9,13].

Over the past decade, ample information has been collected on the brain’s neurogenic capacity in various fish species [15,16], including the comprehensively studied zebrafish [9,17,18,19], that provides new valuable models for studying the cellular and molecular features of neurogenesis and reparative regeneration in the adult brain. The high neurogenic capacity of adult zebrafish [1] and medaka [20] can be explained by the numerous niches containing neural stem cells in their brain. Neurogenic zones have been described from the midbrain of zebrafish, including *tectum opticum* and *torus longitudinalis* [9].

According to modern classifications [21], several groups of nuclei are distinguished in the *tegmentum* and *isthmus* of fish, which are similar in terms of connections and functions performed. One of these groups is represented by structures that provide the conduction of impulses to the motor nuclei of the cranial nerves and to the spinal cord, the so-called premotor structures, or suprasegmental motor region [21,22]. Although in lower vertebrates these areas belong to the reticular formation, they, unlike other reticular structures, are clearly represented in the brains of different vertebrates and can be homologized. Studies of the ultrastructural organization of the tegmentum nuclei and neurogenic zones of this region in the fish brain are extremely rare [9,23]. Previously conducted TEM-based studies have shown seven different cell types (type IIa to type VI) distinguished in neurogenic niches of the adult zebrafish pallium, with radial glia (RG)-like (type IIa) cells present on the surface of the ventricles in the pallial niches and with the subpallial niches consisting of layers of elongated type III cells resembling neuroepithelial (NE) cells [1]. It has been found that the cerebellum and optic nerve retain the aNSPCs with features characteristic of early development [7,24,25], although there is no ultrastructural evidence for this.

At the midbrain level, different groups of vertebrates also have other structures associated with the motor regions, but the lack of information on the connections and functional significance of these structures, including ultrastructural data, does not allow for judgments about their homology with topographically similar nuclei of other animals. For example, in many vertebrates, a deep nucleus of the midbrain (*nucleus profundus mesencephali*) has been described that reaches significant sizes in some groups (e.g., in reptiles). According to available data, in sharks, amphibians, and reptiles, it receives the bilateral tectal input and projects to the underlying motor structures [21].

Similarly to the forebrain, active neurogenesis in adult zebrafish is also maintained in the sensory structures of the olfactory bulb, vagus nerve lobes (XL), facial nerve lobes (VIIL), and the *tectum opticum* (TeO) [23,26,27]. However, the ultrastructural composition and regulation of these niches, their properties of neurogenic plasticity, and the relationship between the information-processing capacity of sensory structures and the aNSPC phenotype remain unknown. Studies on cichlids have shown that the changes in cell proliferation rates in sensory and non-sensory niches appear to be related to the social status [28]. Recently, experiments determining the relationship between the social environment and neurogenesis in adult zebrafish in the telencephalon and sensory niches have shown that the latter are influenced by the conditions of social isolation or novelty to the greatest extent, possibly due to differences in the degree of sensory stimuli [23].

In the central nervous system (CNS) of teleosts, aromatase expression has been detected in neurons [29] and RG cells both during embryonic development and in adults [30,31,32,33]. In contrast to teleosts, aromatase is expressed in amphibian brains only in postmitotic neurons and not in RG cells [34]. The aromatase expression in fish RG is involved in neurogenesis during development and in adults [32,33,35]. The CNS of teleosts undergoes continuous growth during adulthood, and RG cells act as progenitor cells. The aromatase expression in the midbrain of growing juvenile Pacific salmon has not been studied to date. However, the aromatase synthesis in the cerebellum of juvenile chum salmon (*Oncorhynchus keta*) is inextricably linked with constitutive neurogenesis [36] and also increases significantly during regeneration. Of interest is the fact that aromatase has also been detected in progenitor cells in the cerebral cortex of developing mice [37]. Developing neurons have been shown to express aromatase and estrogen receptors in many brain regions [38]. The local synthesis of estradiol and subsequent activation of estrogen receptors are involved in neuronal differentiation, influencing cell migration, survival, death, dendritic outgrowth and axogenesis, synaptogenesis, and synaptic plasticity [38,39,40].

The brain lipid-binding protein (BLBP) is one of the molecular markers of aNSPCs in zebrafish [11]. The BLBP is believed to play a key role in lipid accumulation, membrane synthesis, energy production through lipid transport into mitochondria, cholesterol metabolism, and cell signaling [41,42]. The involvement of the BLBP in cell signaling has been shown to be associated with the interaction of fatty acids and transcription factors such as members of the peroxisome proliferator-activated receptor family [43]. In the adult avian brain, the BLBP is detected in the cerebral and cerebellar RG, as well as in astrocytes. Biochemical studies by Xu and co-authors [44] showed that in the CNS, the ligand for the BLBP is the widespread omega-3 polyunsaturated docosahexaenoic acid (DHA), which is, in particular, necessary for the proper development of the retina and nervous system. In addition, some studies suggest that DHA plays a protective role in the brain against age-related impairments and neurodegenerative diseases such as Alzheimer’s disease [45]. DHA can be locally metabolized in the body to form recently described lipids such as resolvins and neuroprotective autacoids with potential anti-inflammatory properties [46].

In the present study, we used TEM and the immunohistochemical labeling of AroB and BLBP to examine the ultrastructural organization of the suprasegmental nuclei of the tegmentum and characterize the cellular composition of the neurogenic niches of the rostro-medial and caudo-lateral tegmentum in juvenile chum salmon. To identify different types of aNSPCs, we used the IHC labeling of the BLBP in the setting of acute traumatic injury of the *medulla oblongata*. In addition, we examined the distribution of AroB in the tegmentum of juvenile chum salmon following traumatic injury to determine whether post-traumatic changes induce a widespread or *torus semicircularis* (TS) niche-specific neurogenic response and whether this affects different stages of neurogenesis in juvenile chum salmon. These experiments aimed to test the hypothesis that the juvenile chum salmon brain, after being exposed to traumatic injury, is capable of neurogenic plasticity, which is related to the cellular composition of the tegmental neurogenic niche.

## 2. Results

The midbrain tegmentum of juvenile chum salmon contains several suprasegmental nuclei, among which the paramedian region is occupied by the nuclei of the oculomotor complex (NIII), the nucleus of the NFLM, and the dorsal tegmental nuclei (DTN). Figure 1A–C show semi-thin sections demonstrating the major nuclear structures of the tegmentum and the periventricular region containing the lateral (LPVZ) (Figure 1A) and medial (MPVZ) (Figure 1B) periventricular zones. The morphometric characteristics of the examined tegmentum nuclei are presented in Table 1.

The vascular network expression could be seen in the form of venules and arterioles with various degrees, and they were labeled where necessary (Figure 1A,C). Large vessels were identified ventrally of LPVZ, along which cell migration patterns were detected (Figure 1C). The LPVZ and MPVZ contained adult neural stem cells (aNSCs) and were neurogenic regions of the tegmentum of juvenile chum salmon. To date, the neurogenic zones of the tegmentum have remained the least studied and characterized regions for fish. In this report, we, for the first time, provide an ultrastructural characterization of these zones in juvenile chum salmon.

### 2.1. Ultrastructural Characteristics of Tegmentum Nuclei in Juvenile O. keta

#### 2.1.1. Nucleus of the III Nerve

The NIII occupied the rostralmost, paramedian position in the tegmentum of juvenile chum salmon. The ultrastructural organization of the NIII in salmonids was not characterized previously. Large interneurons of three types organized as discrete lenticular groups (Figure 2A) were found in the NIII of juvenile chum salmon. In the intercellular matrix, along with neurons, bundles, and single variously oriented fibers with varying degrees of myelination were also found (Figure 2A). In the dorsomedial zone of the NIII, a population of microglia-like cells was observed (Figure 2A) with a characteristic elongated amoeboid morphology and a dense oval nucleus with condensed chromatin. The cytoplasm of microglial cells was homogeneous without visible organelles (Figure 2A).

The soma size of the largest primary interneurons was 21.15 ± 4.41 μm (long axis) and 13.97 ± 0.46 μm (short axis), respectively (Table 1). In such cells, the outline of the nucleus was poorly visible or not fully visualized (Figure 2A). The nuclei of primary interneurons were lightly stained, the chromatin was not condensed, and the number of nucleoli was one to two per cell (Figure 2A).

In primary interneurons, the light cytoplasm, occupying most of the soma, contained a large number of mitochondria (MT), varying in size and density. The smallest, rounded MT, 208.2 ± 17.8 nm in size, constituted the major proportion (Figure 2B). Larger, elongated MT, measuring 412.2 ± 31.88 nm, account for about 30%. The greatest number of elongated MT were concentrated in the region of the primary dendrite (Figure 2B), where thin actin neurofibrils were also identified. Numerous lipid droplets (Figure 2B), measuring from 300 nm to 500 nm (384.06 ± 71.43 nm), were found in the cytoplasm of primary neurons. Elongated MT and numerous fibrils were localized in the proximal region of the dendrite (Figure 2B).

The soma parameters of secondary NIII interneurons were 17.29 ± 2.63 μm (long axis) and 8.4 ± 1.01 μm (short axis) (Figure 2A,C, Table 1). Nuclei of these cells were usually light and had oval or elongated outlines with weakly condensed chromatin (Figure 2C). The sizes of the nuclei were 8.69 ± 0.93 (long axis) and 5.45 ± 0.63 μm (short axis). In the cytoplasm of secondary neurons, the number of MT were significantly lower than in primary neurons and were dominated by small, rounded MT with branched cristae (Figure 2C,D). In some cases, cisterns of the Golgi apparatus were found near the NIII nucleus, with numerous vacuoles formed near the cis-pole of the latter (Figure 2D). In the cytoplasm, numerous ribosomes were localized on the elongated tubular cisterns of the rough endoplasmic reticulum (RER), the number of lipid droplets were lower than in primary neurons, but sizes of lipid droplets reached 560 nm (Figure 2D).

The smallest interneurons of 9.47 ± 0.39 and 5.33 ± 0.27 μm in size (long and short axes, respectively) were assigned to type III (Figure 2A, Table 1). These cells, as a rule, had elongated nuclei with denser chromatin and a small volume of cytoplasm (Figure 2A). Organelles in the NIII were poorly developed compared to secondary and primary interneurons (Figure 2A).

#### 2.1.2. Nucleus of Fasciculus Longitudinalis Medialis (NFLM)

The NFLM is a premotor nucleus of the basal mesencephalon reticular formation [21]. Inputs from various sources, including the vestibular complex, tectum, and pretectal region, converge on the NFLM neurons [21]. The efferents of the NFLM form part of the *fasciculus longitudinalis medialis* (FLM), which is a multicomponent and multisynaptic pathway common to brains of all vertebrates. The FLM connects the brainstem with the motor regions of the spinal level, is characterized by a constant location, and occupies a paramedian position in the brainstem in juvenile chum salmon, extending over a long distance in the *ventral funiculus* of the spinal cord. The multitude of connections determines the involvement of the FLM in the combination of head and eye rotation [21].

In juvenile chum salmon, the ipsilateral and contralateral parts of the NFLM were represented by heterogeneous groups of neurons (Figure 3A,B, Table 1). The NFLM contained large interneurons with soma sizes of 15.72 ± 0.83 of 8.96 ± 1.21 μm (long and short axes, respectively) (Figure 3A,B, Table 1). The volume of cytoplasm in large neurons was 70–80%. The cytoplasm was light gray or gray, of medium density, and the cells were characterized by the presence of a large number of MTs and lipid droplets (Figure 3A,B, Table 1). A noteworthy feature of the NFLM was the presence of contacts of interneurons with large synaptic endings (Figure 3A–C) containing numerous vesicles with acetylcholine, 35–40 nm in diameter, as well as numerous MT.

Large neurons (INI) had one or two elongated nuclei, measuring 15.72 ± 0.83 and 8.96 ± 1.21 μm (long and short axes, respectively), with moderately condensed chromatin (Figure 3A,C). In the area of the synaptic contact of INI, tight junctions of the desmosome type were observed (Figure 3D,E), as well as all components of the synaptic complex: presynaptic membrane, synaptic cleft, and postsynaptic membrane (Figure 3E). In the presynaptic area, numerous MTs, neurofilaments, and RER cisterns with polyribosomes were found (Figure 3F). The soma sizes in medium-sized neurons (INII) were 11.41 ± 1.61 and 6.02 ± 0.91 µm (long and short axes, respectively), and the cytoplasm in such cells occupied 50–60% of volume (Figure 3A,B, Table 1). The degree of chromatin condensation in medium-sized neurons was higher than in large neurons, with one or more nucleoli visualized (Figure 3B, Table 1). The long axis of the nucleus in INII was 6.89 ± 0.96 µm, and the short axis was 3.55 ± 0.52 µm. The cytoplasm composition in INII was denser than in large neurons, the number of MTs were reduced, the number of lipid droplets were also lower than in large neurons, but their sizes were generally higher (Figure 3B, Table 1).

Along with neurons, oligodendroglia cells were found in the NFLM (Figure 3A,B), as well as single multidirectional myelin fibers and their bundles (Figure 3A,B). The third type of neurons (INIII) in the NFLM was represented by cells with a high nuclear-cytoplasmic index; the cytoplasm in such cells occupied 20–30% of volume (Figure 3A,B). Such cells with soma sizes of 9.08 ± 0.23 and 6.13 ± 0.22 μm (long and short axes, respectively) had a narrow rim of light granular cytoplasm where organelles were almost absent (Figure 3A, Table 1). The INIII nuclei had a high content of heterochromatin and one or more nucleoli (Figure 3A). The presence of neurons with signs of apoptosis in the ipsi- and contralateral parts of the NFLM (Figure 3A,B) were an interesting finding. Late stages of cell apoptosis with pronounced signs of nuclear fragmentation and cytoplasmic pyknosis were also observed (Figure 3B).

#### 2.1.3. Dorsal Tegmental Nuclei (DTN)

Dorsal tegmental nuclei (DTN) were clusters of large neurons in the dorsal and dorsolateral parts of the tegmentum (Figure 4A,B, Table 1). In juvenile chum salmon, DTN were represented by single large neurons with a multipolar morphology (Figure 4A). In such cells, the nucleus was not visualized, the cytoplasm had a granular appearance, the soma sizes were 20.54 (long axis) and 13.33 μm (short axis), three primary dendrites were extended from the cell body, and lipid inclusions were present in the cell (Figure 4A). Neurons with large oval somata (27.95 and 15.89 μm) were located at different depths in the parenchyma of the dorsomedial tegmentum (Figure 4B) and were characterized by the presence of lipid inclusions in the peripheral layers of the cytoplasm. Such cells were surrounded by conducting fibers of the NFLM and ascending thalamic tracts, as well as by oligodendrocytes, astrocytes, and smaller neurons (Figure 4B). Other large neurons (INI) of the DTN (40.34 and 17.95 μm) contained numerous MTs of different sizes and dense bodies in the cytoplasm, a small, elongated nucleus (10.31 and 7.29 μm) with decondensed chromatin, and a dense nucleolus (Figure 4C). Such neurons were in contact with protoplasmic astrocytes and grouped into a small cluster (Figure 4C).

Neuroglial, somato-somatic tight junctions of large DTN neurons and astrocytes were found (Figure 4D, Table 1). Protoplasmic astrocytes with a dense, condensed cytoplasm and short, branched processes ending with characteristic so-called astrocytic feet were identified in the cluster (Figure 4E, Table 2). In some cases, the astrocytic feet were in contact with neuronal fragments (Figure 4F). In general, DTN were represented by a heterogeneous population of neurons, also including smaller INII cells (Figure 4G, Table 1). Such neurons contained large, elongated nuclei with heterochromatin and nucleoli; the degree of chromatin condensation in the neurons varied (Figure 4G, Table 1). MTs and lipid inclusions were localized in the dense cytoplasm of the neurons (Figure 4G). Degraded neurons and apoptotic bodies were identified among the INII population (Figure 4G). Apoptotic patterns of larger INI cells were identified in DTN (Figure 4H). Large apoptotic INI cells containing a fragmented nucleus and cytoplasm, as well as a pyknotic apoptotic body, were located near a neurosecretory cell excreting secretory granules into the intercellular space (Figure 4H). Such features of the DTN ultrastructure indicated the complex mechanisms of regulation of neurons’ functional activity and the neuro–glial relationships in the mesencephalic tegmentum.

Other forms of neuro–glial contacts on the surface of large DTN neurons are shown in Figure 5A. In contrast to the above-described forms, some neurons had an electron-dense cytoplasm with lipid inclusions and a well-defined dense nucleus with several nucleoli (Figure 5B, Table 1). The astrocytic feet on the surface of such neurons had different morphologies, such as thin elongated astrocytic feet (indicated by blue arrow in Figure 5C) and thickened astrocyte feet (red arrow in Figure 5C), which formed tight junctions on the neuron soma (Figure 5D). Desmosome-like structures were revealed at the sites of neuro–glial contacts (Figure 5D). In DTN neurons with a deeper parenchymatous localization, contacts with astroglia were also revealed (Figure 5E). These neurons, measuring 47.95 ± 5.3 μm and 22.24 ± 2.59 μm (long and short axes, respectively), had a less dense cytoplasm with numerous MTs, lipid droplets, and relatively small nuclei displaced toward one of the cell poles (Figure 5E). The chromatin in the nuclei was decondensed and the nucleolus was clearly visible (Figure 5E, Table 1). Neurons contacted large protoplasmic astrocytes (Figure 5E,F), either through thickened end feet (Figure 5E) or through somato-somatic contacts (Figure 5F).

### 2.2. Ultrastructural Characterization of Neurogenic Zones of the Tegmentum of Juvenile Chum Salmon, O. keta

We performed an ultrastructural morphological study to compare the cytoarchitectonic compositions of the rostral tegmental, latero-caudal tegmental zones, and periventricular zone of the *torus semicircularis* (TS) in juvenile chum salmon. Figure 6A–F show semi-thin sections of the tegmental and tectal neurogenic zones that were identified in the juvenile chum salmon examined (Figure 6A–F). All zones were dorsally adjacent to the tectal ventricle (TeV) along the rostro-caudal axis of the midbrain and were composed of several different cell types. Ventrally of the tectal ventricle, the deeper layers of each zone terminated in one or more layers of neurons or adjacent parenchyma (Figure 6A–C). The first complete cell layer, composed exclusively of neurons, was considered the lateralmost boundary of each zone and was not included in the periventricular zone (PVZ). The PVZs were examined at the rostro-caudal transverse levels of the tegmentum to study the ultrastructural organization and cell types comprising each niche: rostral lateral tegmental neurogenic zone (RLTNZ) (×100), TS neurogenic zone (TSNZ) (×200), and caudo-lateral tegmental neurogenic zone (CLTNZ) (×100). A brief description of the characteristic features of the different cell types comprising the PVZ is given in Table 2. When multiple cell types shared some, but not all, of the common features, the cells were divided into subtypes. When cell types could not be determined but were present in a niche, they were classified as unknown.

#### 2.2.1. Rostral Lateral Tegmental Neurogenic Zone (RLTNZ)

The rostral lateral tegmental neurogenic zone (RLTNZ) was identified at the level of the lateral population of the DTN and NFLM (Figure 1A,C and Figure 6A), dorsally adjacent to the tectal ventricle. The majority of cells that comprised the RLTNZ were phenotypically represented by aNSPCs (Figure 7A). Several cell types were found in the RLTNZ of juvenile chum salmon according to the classification of Lindsey et al. [1]. The RLTNZ was divided into two regions separated by a basal membrane (outlined by the black dotted line in Figure 7A). An epithelial sheath, referred to as the dorsal ependymal lamina (DEL), consisting in part of a pseudo-monolayer of ependymal cells (filled in pink) lining the lumen, covered the RLTNZ dorso-laterally and ended at the TeV (Figure 7A). Ependymal cells were typically cuboidal or weakly cuboidal and contained heterochromatin aggregations throughout the nucleus, which was elliptical to elongated in shape (Figure 7A, Table 2). The RLTNZ niche consisted of three or four layers composed primarily of neurons with a small number of non-neuronal cells with various phenotypes (Figure 6A).

In the dorso-medial part of the niche, where the ventricular lumen began, the number of cell layers and diversity of cell morphology increased (Figure 7A). The majority of cells in the RLTNZ were type III and IVa cells, distributed with a high density, and together occupying more than six cell layers deep in the tegmentum (Figure 7A,B). However, elongated ependymal cells were detected in the adjacent tectal marginal zone (TMZ) (Figure 7A, Table 2).

An interesting finding in the RLTNZ of juvenile chum salmon was neuroepithelial-like cells (NECs) (Figure 7B,C). The presence of NECs in the RLTNZ distinguishes juvenile chum salmon from zebrafish and other species and provides TEM evidence for the embryonization phenomenon in the Pacific salmon brain. Studies at higher magnifications showed that NECs can be in a proliferative state (Figure 7C, Table 2), in particular, containing two nuclei under a single membrane.

Along most of the RLTNZ, a segregation of type III cells with irregular and elongated nuclei could be observed, which were characterized by uniformly distributed chromatin, one or two visible nucleoli, and minimal cytoplasm containing few organelles (Figure 7A,B, Table 2). In some areas of the RLTNZ, type IVa cells were visible, with large nuclei containing heterochromatin and lacking nucleoli (Figure 7B,C, Table 2). An ultrastructural examination of the RLTNZ showed that this niche mainly contained type III and IVa cells (Figure 7C–F). The population of type III cells was represented by dark-stained cells with elongated nuclei (Figure 7A,C, Table 2) and cells with irregular nuclei (Figure 7C–E, Table 2). Moreover, many of the type III cells were oriented parallel to the TeV (Figure 7A). Type IVa cells in the RLTNZ were lightly stained cells with heterochromatin and a narrow rim of cytoplasm (Figure 7C–F, Table 2). Type IVa cells were often characterized by irregularly shaped nuclei with varying degrees of dark staining, reticulated chromatin, and scanty cytoplasm containing few organelles (Figure 7C–E). Type IVa cells typically formed clusters (Figure 7B,C). Adult-type glial cell division patterns were detected in the lateral regions of the RLTNZ (Figure 7B).

#### 2.2.2. *Torus Semicircularis* Neurogenic Zone (TSNZ)

The *torus semicircularis* neurogenic zone (TSNZ) was one of the largest neurogenic zones of the tegmentum in juvenile chum salmon that occupied most of the TS (Figure 6B). In the dorsomedial part of the niche, near the intertectal ventricle lumen, the number of cell layers was six to seven (Figure 8A, Table 2). In the lateral part of the TSNZ, the number of cell layers increased to 12–14 and formed characteristic thickenings (Figure 6C,D). During the ultrastructural analysis, the medial part of the TSNZ was studied in the most detail; the greatest morphological heterogeneity of cell types was observed in this niche (Table 2). The TS is an analogue of the posterior pair of the *quadrigemina* of the midbrain in higher vertebrates [1].

Cells within the medial part of the TS neurogenic zone (MTSNZ) were arranged into several rows toward the edge of the niche. These cells were oriented tangentially to the TeV with distinctly elongated outlines of nuclei (Figure 8A), which demonstrated potential features of this cell type. In some cases, one pole of the nucleus of type III cells was in contact with the apical membrane at the ventricular border, while the opposite end was located basally, regardless of whether the cells were oriented perpendicular or parallel to the ventricle (Figure 8B). In contrast to type III cells in neurogenic niches of the adult zebrafish forebrain [1], these cells in juvenile chum salmon were adjacent with neighbor cells of the same morphology and the more deeply located type IVa cells (Figure 8B). The population of type III cells was represented by dark-stained cells with elongated nuclei (Figure 8B,C) and cells with irregular nuclei (Figure 8B–D). Moreover, many of the type III cells, unlike those in the RLTNZ, were oriented perpendicular to the tectal ventricle (Figure 8A,C). Type III cells in the TSNZ were characterized by uniformly distributed chromatin, with several condensed areas and minimal cytoplasm, often shifted to one of the cell poles, and contained several organelles and lipid droplets (Figure 8C,D, Table 2).

Microvessels were occasionally found in the TSNZ whose wall was composed of endothelial cells and immature forms of macrophages containing a nucleus and branched cytoplasmic processes localized in the central part of the wall (Figure 8E). The peripheral cytoplasm of such cells showed signs of stratification and vacuolization, thus indicating a state of apoptosis (Figure 8E). However, the most abundant cell type in the TSNZ was type III cells with irregularly shaped nuclei and a minimal volume of cytoplasm adjacent with the intercellular substance (Figure 8F).

#### 2.2.3. Caudo-Lateral Tegmental Neurogenic Zone (CLTNZ)

The CLTNZ niche was located latero-caudally deep in the periventricular layer of the TeO, adjacent to the TeV and the periventricular layer of the tegmentum (Figure 6E,F). This niche consisted of both type III and IVa cells, which together occupied more than six cell layers in depth (Figure 6E,F, Table 2). At the caudal border, where the CLTNZ cells contacted cells of the periventricular TeO, the thickness of the CLTNZ doubled (Figure 6F). The CLTNZ cells were surrounded by an intercellular substance where single vessels were also identified (Figure 6F).

Type III cells with irregular and elongated dark-stained nuclei and narrow rims of cytoplasm were located along most of the niche boundary; among them, type IVa cells were noticeable (Figure 9A, Table 2). The latter cells were more frequently grouped into small clusters and had lighter nuclei with nucleoli surrounded by scanty cytoplasm (Figure 9A, Table 2). The population of type III cells in the CLTNZ was often localized parallel, and less often perpendicular, to the TeV with clearly elongated outlines of nuclei (Figure 9A), which demonstrated the potential features of such cells. In the CLTNZ, these cells primarily contacted neighboring cells of the same morphology and more deeply located type IVa cells (Figure 9A). The morphological phenotypes of type III and IVa cells are shown in Figure 9B.

In the caudal regions, the CLTNZ had noticeable thickenings; in this case, the number of cell rows in the niche was seven to eight (Figure 9C). In these areas, type III cells dominated, whose nuclei had irregular and elongated morphologies (Figure 9C,D, Table 2). In this part of the CLTNZ, type III cells were localized perpendicular to the TeV (Figure 9C). The number of cells with elongated nuclei was much higher at the niche surface, whereas cells with irregularly shaped nuclei were localized in deeper layers (Figure 9C), where type IVa cells were also found (Figure 9C,D). In the medial part of the CLTNZ, the cell distribution density was moderate to high, dominated by type III cells, and the niche thickness in this zone was six cells (Figure 9E). In the apical layers, patterns of type III cell proliferation were common (Figure 9F). In such cells, two nuclei were observed under a common plasma membrane without the separation of the cytoplasm (Figure 9F).

### 2.3. Immunohistochemical Labeling of BLBP

#### 2.3.1. Immunohistochemical Labeling of BLBP in the Tegmentum of Intact Fish

Previously, the BLBP was shown to label RG cells in adult and larval zebrafish [26,47,48,49,50]. To elucidate the pattern of BLBP expression in the mesencephalic tegmentum of juvenile chum salmon, previously described from adult human and zebrafish brains, we decided to additionally document its expression using BLBP immunohistochemistry.

In the rostral part of the tegmentum of intact juvenile chum salmon, the BLBP expression in the periventricular RLTNZ was quite sparse (Figure 10A). No BLBP expression was recorded directly from the RLTNZ; in the medial part of the projection, expression patterns were observed in the subventricular zone (SVZ) in small, elongated type I cells [51], in deeper, parenchymatous layers of the lateral tegmentum, and in parenchymatous areas (Figure 10A). In general, the BLBP expression was low in intact juvenile chum salmon.

In more caudally located areas of the tegmentum, at the level of the isthmus, in the rostral projections of the *valvula cerebelli*, the BLBP expression was detected in small type 1 cells located in the molecular layer (ML); in some cases, a polynuclear pattern of BLBP expression in cells was observed (Figure 10B).

At the level of the NIII paramedian projection, the BLBP expression was detected in single cells (Figure 10C). There were also small clusters of various densities consisting of small cells, as well as patterns of cell migration along vessels (Figure 10C). In the caudo-lateral direction, similar patterns of cell migration along guides, clusters of small cells, and cell bodies of reticulospinal neurons were observed; single cells expressing the BLBP were also detected (Figure 10D). In the TS region, single BLBP-ip cells and large and dense clusters consisting of BLBP-immunonegative undifferentiated cells of an unknown nature were detected (Figure 10E). In the ventral part of the TS, single cells with BLBP-expressing nuclei were detected (Figure 10F), whose immunolabeling intensity varied from moderate to high.

#### 2.3.2. Immunohistochemical Labeling of BLBP in the Tegmentum After Traumatic Injury

As a result of acute traumatic injury to the *medulla oblongata* of juvenile chum salmon, the patterns of BLBP expression in the cells and RG fibers in the mesencephalic tegmentum and neurogenic zones of the tegmentum significantly changed. The study was conducted on day 3 post-injury.

In the rostro-lateral parenchymatous region of the tegmentum, an extensive aggregation of small, undifferentiated immunonegative cells surrounded by patterns of cell migration were found (Figure 11A). Elongated, intensely BLBP-ip cells migrated, forming tracks, including those along small vessels, and diffuse clusters of undifferentiated cells around them were also detected (Figure 11A). Single BLBP-expressing cells were also observed in the parenchyma of the tegmentum (Figure 11A). The morphometric characteristics of BLBP-ip cells are presented in Table 3.

In the medial tegmentum, patterns of migrating immunonegative undifferentiated cells were also present (Figure 11B). Along with elongated, intensely labeled BLBP-ip cells, the migration of intensely labeled rounded or oval BLBP-ip cells along small vessels were a characteristic pattern (Figure 11B). In some cases, multicellular aggregations were found, including immunonegative cells, elongated BLBP-ip cells, and BLBP-ip fibers (Figure 11B). The morphometric characteristics of migrating BLBP-ip cells of rounded/oval shapes are presented in Table 3.

During the acute post-traumatic period, in the caudal zone of the tegmentum, BLBP-ip Bergmann glia fibers were found, located in the dorsal part of the molecular layer (ML) in the region of the *valvula cerebelli* (Figure 11C). The migration of immunonegative cells was observed along the tortuous BLBP-ip Bergmann glia fibers and diffuse aggregations of similar cells were observed in the upper third of the ML (Figure 11C). Terminal branches of immunopositive Bergmann glia had either simple, branched end apparatuses or typical end feet (Figure 11D inset). In deeper layers of the ML, clusters of larger, intensely labeled oval and rounded BLBP-ip cells were identified (Figure 11C), located in the lower third of the ML along with diffuse clusters of immunonegative small cells (Figure 11D).

#### 2.3.3. BLBP Expression in RG of the Tegmentum Post-Injury

A characteristic feature of post-traumatic BLBP expression was its localization in RG fibers (Figure 12A–E). BLBP-ip patterns of RG fibers in the acute post-traumatic period were detected in all anatomical areas of the tegmentum.

BLBP immunolabeling patterns were identified in RG fibers ventro-medially of the NFLM (Figure 12A). RG fibers with BLBP immunopositivity in the tegmentum formed a network and resembled BLBP-ip Bergmann glia fibers in the *valvula cerebelli*, but in the tegmentum, BLBP-ip RG fibers were more extensive (Figure 12A), with immunonegative cells migrating along them. In the parenchyma, diffuse clusters of small immunonegative cells and larger, moderately labeled oval BLBP-ip cells were found between the BLBP-ip fibers of RG (Figure 12A).

Thick BLBP-immunolabeled guides containing BLBP-ip cells from the neurogenic zone of the TS extended from the periventricular zone (PVZ) to the TS region (Figure 12B). In the ventro-medial part of the TS, BLBP+ RG fibers and their end feet were found, as well as multicellular complexes with BLBP+ and BLBP− cells (Figure 12B), but BLBP-expressing fibers formed a less dense network than in the NFLM region. In the lateral torus (LT) region, BLBP-ip cells migrating from the neurogenic zone were more common than TS, whereas BLBP-ip RG fibers were single (Figure 12C). In the dorsolateral and lateral marginal regions of the LT, clusters of intensely labeled BLBP-ip aNSPCs were found (Figure 12C). In the LT, massive patterns of BLBP-ip cell migration were detected, between which moderately oval BLBP-ip cells were identified (Figure 12C).

In the dorsal tegmentum, moderate BLBP expression was detected in large DTN neurons (Figure 12D). In the basal part of the tegmentum, BLBP-ip RG fibers with their end apparatuses were present, as well as single, intensely labeled, rounded BLBP-ip cells localized between RG fibers (Figure 12D). The morphometric parameters of these structures are given in Table 3. In the caudal parts of the tegmentum, intensely labeled migrating BLBP-ip cells and single BLBP-ip cells dominated the basal and medial regions (Figure 12E). In the ventrolateral part of the tegmentum, intensely labeled BLBP-ip cells were identified in the marginal and submarginal regions (Figure 12F, Table 3).

#### 2.3.4. BLBP Expression in Neurogenic Zones of the Tegmentum After Acute Traumatic Injury

After traumatic injury, a high induction of BLBP expression in aNSPCs was detected in the neurogenic zones of the tegmentum of juvenile chum salmon (Figure 13A). In the RLTNZ, BLBP expression was detected in aNSPCs according to Lindsey’s classification [1], arranged into six rows in the PVZ of the tegmentum at the level of NFLM and DTN (Figure 13A). Intensively labeled BLBP-ip aNSPCs were found in the subependymal zone of the RLTNZ, but radial guides extended in the ventrolateral direction, with patterns of BLBP-ip aNSPC migration in different directions identified within them (Figure 13A). In the ventral direction of the tegmentum, in particular, an increased distribution density of migratory cell flows was observed, indicated by fragments of such tracks (Figure 13A, outlined by the red square). In the lateral direction, vice versa, there was an increased number of BLBP-ip RG fibers (Figure 13B, red square). The appearance of BLBP-ip aNSPCs in the RLTNZ along with immunonegative cells, as well as the activation of cell migration in the post-traumatic period (Figure 13C), also show the involvement of BLBP in the post-traumatic process. In the parenchymatous zone adjacent to the RLTNZ, other parts of aNSPCs were also found: RG fibers and their terminal apparatuses (Figure 13D). The revealed migration patterns indicate intensive post-traumatic processes associated with the mass proliferation and migration of aNSPCs to the injury zone. The BLBP in the tegmentum of juvenile chum salmon is, thus, an important participant in post-traumatic processes such as the maintenance, proliferation, and differentiation of neural stem cells, which is consistent with previous data [52,53,54,55]. The results suggest that BLBPs are indeed important factors not only for brain development and neurogenesis but also in post-traumatic neurogenesis.

### 2.4. Immunohistochemical Labeling of Aromatase B (AroB)

#### 2.4.1. Immunohistochemical Labeling of AroB in the Tegmentum of Intact Fish

In the fish brain, AroB is expressed in RG cells [30,47,56]; this is in contrast to mammals, in which AroB is expressed in neurons and reactive astrocytes post-injury [57].

Moderate AroB expression in the tegmentum of juvenile chum salmon was detected in single, small, rounded type I cells in the SVZ in the dorsomedial and dorsal parenchymatous regions (Figure 14A, Table 4). Single larger cells with a higher immunolabeling intensity were detected in the medio-basal parts of the tegmentum (Figure 14A). In the dorsolateral tegmentum, no AroB immunolabeling patterns were detected in cells (Figure 14B, Table 4), but multidirectional migration patterns of immunonegative cells were observed (Figure 14B, yellow arrows).

In the rostral part of the TS, single granule-like AroB expression was detected in small cells of the SVZ (Figure 14C). In deeper layers of the parenchyma, single oval-shaped cells weakly expressing AroB were found (Figure 14C, Table 4). Mass migration patterns of immunonegative cells were observed in the TS (Figure 14C). In the NIII region, intense AroB expression was detected in cells of the medio-basal region (Figure 14D, black rectangle). Single small cells weakly labeled with AroB were detected in the parenchyma (Figure 14D).

In the caudal TS, weak AroB expression patterns were identified in small, rounded cells of the SVZ (Figure 14E, red arrows) and more intense patterns in granule-like cells (Figure 14E, red rectangle). Patterns of multidirectional migration of immunonegative cells were observed mainly in the parenchymatous part of the TS (Figure 14E, Table 4). In the ventro-lateral part of the isthmus, single, moderately and intensely labeled AroB+ cells (Figure 14F, Table 4) and single immunonegative neurogenic niches were identified (Figure 14F, red oval).

#### 2.4.2. Immunohistochemical Labeling of AroB in the Tegmentum After Traumatic Injury

Following acute brainstem injury, maximal AroB expression was observed at the isthmus level of the juvenile chum salmon brain, both in the brainstem and in the *valvula cerebelli*.

#### 2.4.3. Immunohistochemical Labeling of AroB in the Rostral Isthmus

In the rostral isthmus (Is), AroB induction was detected in populations of cells migrating along the guide cells from the basal and basolateral regions of the brain (Figure 15A). In the trigeminal ganglion and pial membrane, several types of AroB+ cells were identified: large, moderately labeled cells (Figure 15A, yellow arrows, Table 4); small, intensely labeled cells (Figure 15A, red arrows, Table 4); and mixed clusters of large and small intensely labeled cells (Figure 15A, red dotted square, Table 4). In the basal and basolateral part of the isthmus, AroB+ RG fibers were identified (Figure 15A, orange arrows). In the parenchymatous areas of the isthmus, single, intensely labeled AroB+ cells of a medium size were identified (Figure 15A, green arrows, Table 4). In the PVZ of the isthmus, the AroB immunoreactivity was low, with single, small, intensely immunolabeled cells encountered (Figure 15A, red square, Table 4). Single AroB+ cells of a similar type were found in the granular layer of the *valvula cerebelli* (Figure 15A, black square, Table 4).

It was found that small, intensely labeled AroB+ cells, along with immunonegative cells, migrate tangentially from unknown peripheral sources localized at the border with the trigeminal ganglion (Figure 15B). AroB+ RG fibers with characteristic end feet were found within these complexes (Figure 15B). Some of the AroB+ fiber complexes were located within the trigeminal ganglion (Figure 15A,B). Similar structures were also found in more rostrally located parts of the isthmus (Figure 15C). Dense local patterns of migration of intensely labeled AroB+ cells in the ventrolateral regions of the tegmentum (Figure 15C, red inset) were combined with AroB+ RG fibers whose end feet converged on immunonegative cells (Figure 15C, orange arrows). Migrating AroB+ cells were located along the immunopositive RG fibers (Figure 15C, Table 4). In the ventro-medial part of the tegmentum, more extended fragments of RG fibers were found, along which AroB+ and AroB− cells migrated (Figure 15D). Such fibers were directed ventro-laterally and ventro-medially, with their distribution density slightly lower than in the basolateral part of the tegmentum (Figure 15A,D). Scattered clusters of immunonegative cells were found between the RG fibers (Figure 15D, red dotted oval).

In the baso-medial part of the tegmentum, the density of RG fibers increased (Figure 15E). The patterns of cell migration along the RG fibers were less organized in the radial direction than in the lateral part of the tegmentum (Figure 15E), although the morphology of AroB+ cells migrating along fibers was similar. In the medial part of the tegmentum, in addition to longitudinal migratory AroB+ cell complexes, dense clusters of small, intensely labeled AroB+ cells with a parenchymatous localization were detected (Figure 15E, red dotted rectangle). We identified these dense clusters as secondary reactive post-traumatic neurogenic niches. Similar niches were detected in the outer marginal zone of the tegmentum, at the border with the pial membrane (Figure 15F, red dotted rectangle). In the border areas, not only AroB+ RG fibers but also microvessels guided the migration of AroB+ cells (Figure 15F). Thus, after traumatic injury in the basolateral and baso-medial tegmentum of juvenile chum salmon, AroB expression was detected in different cell types and RG, whereas such patterns were not observed in the PVZ and SVZ. In the PVZ and SVZ during the acute post-traumatic period, low AroB activity was recorded from single, small rounded cells of the PVZ (Figure 15A), while patterns of the radial migration of AroB+ cells and RG in the PVZ and SVZ were not detected.

#### 2.4.4. Immunohistochemical Labeling of AroB in the Caudal Part of the Isthmus

In the caudal zone of the isthmus, the distribution density of AroB+ cells and RG fibers was higher than in the rostral zone (Figure 16A). In the caudal part of the isthmus, an asymmetric distribution of AroB+ cells and RG fibers was recorded in the post-traumatic period (Figure 16A). On the one hand, a high density of distribution of AroB+ complexes, including cells and fibers of the RG, was identified in the ventro-medial region (Figure 16A).

The immunolocalization of AroB in DTN neurons after injury was not detected (Figure 16A). Single AroB+ cells were detected in the PVZ, and moderately labeled RG fibers were present in the SVZ (Figure 16A). Dorsally of the NFLM, patterns of AroB+ cell migration along the RG were revealed on the contra- and ipsilateral sides (Figure 16A).

In the medio-basal part, a secondary post-traumatic niche of AroB+ cells was detected (Figure 16A,B, red dotted rectangle). Along the lateral wall of the infundibulum on the ipsilateral side, a dense linear cluster of AroB+ cells and single intensely labeled AroB+ cells were found (Figure 16A,B, red square). On the contralateral side, the density of AroB+ RG fibers was high in the ventro-medial region (Figure 16A). These fibers converged into dense, large, and smaller post-traumatic neurogenic niches containing AroB+ cells (Figure 16A,B, red dotted rectangles). More dorsally, a linear cluster of AroB+ cells distant from the III ventricle and single intensely labeled AroB+ cells similar to those on the ipsilateral side were found (Figure 16C, red inset). Intensely immunolabeled AroB+ RG, end apparatuses, and fiber fragments were detected on the ipsilateral side (Figure 16D). However, in the ventro-lateral tegmentum on the contralateral side, the distribution density and intensity of AroB+ cell RG complex immunolabeling were lower than on the ipsilateral side (Figure 16E). Multidirectional patterns of AroB+ RG and their complex interactions with migrating cells were found throughout the ventro-medial tegmentum of juvenile chum salmon on the ipsilateral side (Figure 16F).

#### 2.4.5. Immunohistochemical Labeling of AroB in the Valvula Cerebelli

As a result of acute traumatic injury, significant AroB+ expression was detected in the *valvula cerebelli* located in the intertectal lumen at the level of the isthmus. At the caudal level, the number of AroB+ cells and fibers in the central part of the molecular layer (ML) were limited (Figure 17A). Migrating AroB+ cells were detected along the radially oriented Bergmann glia fibers (Figure 17A). Small clusters of such cells were identified near the lumen of the cerebellar ventricle (Figure 17A).

In contrast, in the rostral part of the *valvula cerebelli*, massive patterns of AroB+ cell migration from the central region of the ML were recorded (Figure 17B). Along with immunopositive cells, large clusters of small immunonegative cells were observed (Figure 17B). The guides, along which AroB+ cells migrated, were represented by microvessels containing immunonegative endothelial cells (Figure 17C). The largest, centrally located complex of vessels was found, with dense clusters of AroB+ cells on its surface (Figure 17C). Smaller vascular structures were detected near the lumen of the cerebellar ventricle, along which clusters of migrating AroB+ cells were localized (Figure 17D). In the peripheral zones of the *valvula cerebelli*, patterns of AroB+ cell migration and fragments of AroB+ Bergmann glia were identified in the ML (Figure 17E).

## 3. Discussion

The conducted studies allowed us to identify, for the first time, the ultrastructural organization of the suprasegmental nuclei in the tegmentum of juvenile Pacific chum salmon, which has never been performed before for salmonids. As a result of the ultrastructural analysis of three large nuclei (the NIII, NFLM, and DTN), we identified the features of the neural organization of these previously uncharacterized structures of the tegmentum (details are summarized in Table 1). In particular, we divided neurons in the NIII, NFLM, and DTN into three major groups differing in sizes of the soma, nuclei, and in cytoplasmic characteristics (Table 1). In the course of this study, we revealed the differences and similarities in the organization of heterogeneous interneuron groups forming the structure of the tegmentum nuclei that determine the basic features of their morphological and ultrastructural organization.

### 3.1. Features of the Ultrastructural Organization of the Tegmentum Nuclei in Juvenile Chum Salmon

The results of neurochemical studies on salmonids such as brown trout and rainbow trout [58], masu salmon *Oncorhynchus masou* [59], and zebrafish [60,61] have shown that the motor nuclei of the tegmentum, in particular NFLM, DTN, and NIII, are cholinergic and express choline acetyltransferase (ChAT) [58,59]. Data on zebrafish turned out to be more heterogeneous. In particular, as Clemente et al. showed in 2004 [60], ChAT immunoreactivity in zebrafish was confined to the rostral segmental, oculomotor, and acetabular nuclei within the mesencephalic segment; a wide distribution of acetylcholinesterase (AChE) reactivity was also observed in this region [60]. In the studies by Müller and co-authors [61], ChAT immunoreactivity has been detected in the Edinger–Westphal nucleus, the oculomotor nucleus, and the rostral tegmental nucleus of zebrafish [61]. Among amphibians, ChAT immunopositivity was recorded from the tegmental structures of *Gekko gecko* [62].

The cholinergic neurotransmission in the suprasegmental centers of the juvenile chum salmon tegmentum may explain the distant synaptic neurotransmission carried out by large synaptic formations of the excitatory type, which we identified in the NFLM (Figure 3B–E). The synaptic structures identified in the tegmentum of juveniles were asymmetrical, indicating the excitatory nature of the synapse. Synaptic vesicles were light, rounded, or oval, and 35–40 nm in diameter, which corresponded to the size of vesicles in cholinergic neurotransmission. The presence of large synaptic terminals in the NFLM in juvenile chum salmon explains their functions as a motor integrator, since inputs from multimodal sources, including the vestibular complex, tectum, and pretectal area, converge on NFLM neurons [21]. NFLM efferents form part of the FLM and, in some animals, are directed toward the nuclei of the oculomotor complex [63]. The FLM is a multicomponent pathway found in the brain of all vertebrates. It connects the brainstem with the motor regions of the spinal cord, is characterized by a constant location, and occupies a paramedian position in the brainstem, passing into the ventral funiculus of the spinal cord over a long distance. The multiplicity of connections determines the involvement of the FLM in such reactions as the coordinated rotation of the head and eyes [21].

In addition to the synaptic system, there is another system of intercellular communication in the juvenile chum salmon brain, which arises in the early stages of postembryonic development and represents intercellular interactions carried out by the paracrine pathway during the period when cells have not yet developed processes and the synaptic structure. However, such poorly differentiated cell forms are already capable of expressing specific syntheses: some neurotransmitters and enzymes synthesizing them, transcription factors, etc. [64]. We suggest that most signals synthesized at the juvenile ontogeny stages are involved in regulating the differentiation of target neurons and the expression of their specific phenotype.

In the cell populations of NFLM and DTN in juvenile chum salmon, we found cells with signs of apoptosis (Figure 3B and Figure 4G). The causes of apoptosis of neurons in the suprasegmental nuclei of the tegmentum in juvenile chum salmon are not clear. These may result from a physiological utilization, but the question as to why functioning neurons are eliminated in the growing brain of chum salmon remains unanswered. As our ultrastructural studies showed, in the DTN of juvenile chum salmon, in some cases, cells with paracrine neurosecretory activity were localized near large apoptotic neurons (Figure 4H). It is possible that such an effect can induce apoptosis in DTN neurons.

### 3.2. Apoptosis in the Tegmentum of Juvenile Chum Salmon and Other Fish

Apoptosis, as a form of programmed cell death, is the most important component of maintaining homeostasis and growth in all fish tissues, and plays an important role in immunity and cytotoxicity [65]. It is known that in the fish CNS, apoptosis, which is a pure type of cell death, does not induce an inflammatory response [66]. The outer membrane of the cell remains intact, and neighboring cells and tissues are not damaged [67,68,69]. The major morphological signs of apoptosis observed in the NFLM and DTN cells of juvenile chum salmon were the cell shrinkage, cytoplasmic condensation, membrane swelling, nuclear fragmentation, and formation of apoptotic bodies (Figure 3B and Figure 4G). The latter, we suggest, are rapidly and orderly destroyed by microglia or resident macrophages that in juvenile chum salmon are also found among the tegmentum neurons (Figure 2A).

The results of studies on the cerebellum of *Apteronotus leptorhynchus* confirm that one of the main functions of microglia and macrophages may be the removal of remnants of cells that have undergone apoptosis at the site of injury. It has been hypothesized that this pure type of cell death is a major factor in the rapid and, sometimes, complete regeneration of neural tissue [66]. Similar findings are reported in studies of traumatic eye injury in adult rainbow trout (*Oncorhynchus mykiss*) [70] and in studies of the Wallerian degeneration of the optic tract in *Carassius auratus* [71,72,73]. A two-day study following spinal cord transection in zebrafish showed that, in addition to myelin phagocytosis, a macrophage/microglia response was observed caudally of the transection site using immunohistochemistry and electron microscopy [74].

Currently, teleost fishes are the vertebrate group whose immune system provided the most comprehensive information available due to the discovery of genes homologous or orthologous to mammalian immunomodulatory molecules [75]. Comparative genomic studies have found clear unambiguous fish orthologs for only a few members of the mammalian interleukin (IL-1) family; however, the complete genome sequencing of *Fugu rubripes* have revealed an ortholog for fish IL-18 and its putative receptor complex [76].

Although little is known about cell-mediated immunity in teleosts, there is evidence that controlled cell death limits the accumulation of harmful or potentially dangerous cells [77]. In particular, apoptosis is known as an important mechanism of the immune response in fish and an effective antiviral defense strategy [78].

Previous studies showed that the apoptotic program is activated by the cell itself when necessary [67]. This program is genetically controlled in intact animals and results in a physiological response elicited by specific suicide signals or is induced by the absence of prosurvival signals. In fish, many genes were identified that control programmed cell death [79]. The fate of aneuploid cells in the brain of adult brown knifefish, *Apteronotus leptorhynchus* (Gymnotiformes, Teleosts), was investigated to elucidate the issues about their apoptotic elimination [80]. It has been shown that cells lacking prosurvival genes are eliminated by apoptosis not only early in development but also later in ontogenesis, when the consequences of inappropriate gene expression become apparent [80].

Several apoptosis-regulating genes and two death receptors have been identified in zebrafish [81]. The rainbow trout caspase 6 gene has been cloned and sequenced [82]. These findings have been widely discussed with regard to the regulatory role of apoptosis in fish immune responses [65]. The complex pathways by which cell death is induced in the CNS depend, in part, on the nature of the death signal, the tissue type and stage of development, and the physiological environment [67,83,84]. Apoptosis can be induced by cell injury and ultimately by activation of various death receptors that are expressed on the surface of most cells. The intracellular level of adenosine triphosphate (ATP) is also considered to be a critical factor in the choice of death type, as apoptosis is an ATP-dependent process, whereas necrosis is an energy-independent mode of cell death [85,86,87].

Caspases, Bid, Bax, the Bcl-2 family of proteins, and heat shock proteins (HSPs) involved in the regulation of apoptosis have been well studied and documented [88]. Apoptosis has been defined as caspase-mediated death, including Cas2, Cas3, Cas6, Cas7, Cas8, Cas9, and Cas10 [89]. The major pathways of apoptosis have been proposed as the extrinsic or death receptor pathway, the intrinsic or mitochondrial pathway, and the T-cell-mediated granzyme B pathway [67]. In fish, a further distinction has been made between the extrinsic and intrinsic pathways of apoptosis activation. However, current commercial caspase-specific substrates and caspase-specific inhibitors are highly heterogeneous, and results obtained using them should be interpreted with caution [90]. Evidence suggests that certain molecules such as the mitochondrial permeability transition pore (PTP) and the mitochondrial outer membrane permeabilizer (MOMP) may be involved in the apoptotic mechanism in both fish and mammals, but the mechanism of action of these molecules remains largely unknown [90]. The relationship between apoptosis and HSPs also requires further study. When cells are damaged, one of the two opposing responses may occur. The first response is apoptosis that occurs as a result of pathological changes. This response is characterized by significant changes in cellular architecture, leading to self-destruction [90]. The major role of apoptosis is to prevent inflammation by removing damaged cells. The second response is the rapid synthesis of HSPs, which prevents injury or facilitates cell repair to maintain cell viability [91,92,93,94,95]. Both pathways can be induced by a variety of environmental factors and have a significant impact on the biological consequences of stress. Cell fate is determined by the interplay between these pathways [96].

Apoptosis studies often face methodological dilemmas, in particular, the lack of approaches to the intravital imaging of apoptotic cells [97]. As a result, the clear understanding of the behavior of apoptotic cells in living tissue has been elusive. The advent of intravital monitoring methods to track apoptotic cells as they are generated in and cleared from the zebrafish brain has facilitated the discovery of the remarkable motility of apoptotic cells that are capable of moving significant distances. It has been assumed that the death of apoptotic cells in living vertebrates is mediated by the combined action of apoptotic cell migration and the elmo1-dependent macrophage uptake of the guanine nucleotide exchange factor [97].

### 3.3. Neuroglial Interactions in the Tegmentum of Juvenile Chum Salmon

By studying the ultrastructural organization of the tegmental nuclei in juvenile chum salmon, we identified numerous neuro–glial contacts for the first time (Figure 4C–F). The first such contacts were found between large dorsal tegmental neurons in DTN and protoplasmic astrocytes (Figure 4C,D). Since the majority of astrocytic glial cells in the fish brain are represented by radial glia and often retain their radial identity throughout life [98,99,100,101], the protoplasmic-type astrocytes found in the tegmentum of juvenile chum salmon represented the final stage of astrogliogenesis. Such cells lose their neurogenic capacity, and we suggested that they were involved in maintaining the metabolism of large tegmental neurons and also directly affected neuronal connections by forming tight junctions with the neurons of DTN (Figure 5C,D). We also assumed that the cells with dense, dark-stained cytoplasm and several processes, found in the tegmentum of juvenile chum salmon, were astrocytes that had separated from the ventricle and had a typical astrocytic phenotype. Protoplasmic astrocytes were found in DTN and had several main processes that ended in typical end feet (Figure 4E).

Astrocytes are known to express potassium and sodium channels and are able to induce inward currents [102], but, unlike neurons, astrocytes do not propagate action potentials along their processes. However, astrocytes can regulate increases in intracellular Ca^2+^ concentration, which is a form of astrocyte excitability. We suggested that regulated Ca^2+^ currents play a functional role in the established astrocyte–astrocyte and astrocyte–neuron interactions in the juvenile chum salmon DTN. Studies showed that increases in astrocyte Ca^2+^ can initially be induced as intrinsic oscillations resulting from the release of calcium from intracellular stores, but can also be triggered by mediators (including glutamate and purines) released during neuronal activity [103,104]. In addition, increases in Ca^2+^ can induce the release of intercellular messengers such as glutamate from astrocytes into the extracellular space, thereby triggering receptor-mediated currents in neurons and propagating to neighboring astrocytes [105].

In addition, unlike those in mammals, fish astrocytes contain the enzyme glutamine synthetase, which converts toxic glutamate to neutral glutamine [106]. Recent studies on the telencephalon [107] and cerebellum of juvenile chum salmon [36] have shown an increase in GS-expressing astrocytes after traumatic injury.

Astrocytes are known [108] to be involved in synaptic transmission through the regulated release of glutamate, gamma-aminobutyric acid (GABA), and D-serine. As was previously reported for juvenile masu salmon, GABA-ergic neurons are localized in the DTN area [64]. The release of gliotransmitters, accompanied by an increase in Ca^2+^ current in astrocytes, occurs in response to changes in the synaptic activity of neurons [108,109] and can modulate the activity of large GABA-ergic neurons in the tegmentum of juvenile chum salmon. In addition to directly influencing synaptic activity through the release of gliotransmitters, astrocytes have the potential to exert potent and long-lasting effects on synaptic function through the release of growth factors and cytokines [105]. Astrocytes are also sources of neuroactive steroids (neurosteroids), including estradiol, progesterone, and aromatase, which can exert synaptic effects on GABA receptors [57,110,111,112]. The different types of astrocyte–neuron contacts identified in the DTN of juvenile chum salmon (Figure 5C,D) allow for astrocyte activation via neurotransmission, as astrocytes express different types of glutamate receptors [113]. Upon astrocyte activation, signaling pathways involving extracellular signal-regulated kinase (ERK) and c-Jun N-terminal kinase (JNK) are initiated [114].

Another glial type identified in the tegmentum of juvenile chum salmon was microglia (Figure 2A). Microgliocytes are resident macrophages of the brain and are the main immune cells of the CNS that respond rapidly to injury, infection, and inflammation [115]. Microglial cells of the fish optic nerve have been most comprehensively studied in *Carassius auratus*, including at the ultrastructural level [116,117]. Microglial cells of juvenile chum salmon resemble mammalian microglia; they have a heterochromatic nucleus and high levels of extended RER [116]. *Carassius auratus* is also an alternative model for studying the role of microglia in the development of the Wallerian degeneration of the optic nerve [71,72,73,118]. Subsequent studies on fish have described microglial cells, surrounded by supramedullary located neurons, at the ultrastructural level [71,117]. Following CNS injury, quiescent ramified microglia retract their processes, retain the activated amoeboid phenotype, become motile, and migrate to the site of injury [119]. Thus, resident microglial cells in the tegmentum of juvenile chum salmon are a key component of the CNS’s innate immunity to pathogeneses, cellular damage, and toxic proteins. Microglia are involved in this natural response by providing, together with infiltrating macrophages, inflammatory mediators and by participating in neuroprotective and neurodegenerative processes [120].

Immunohistochemical labeling in the tegmental region performed on *Tetraodon fluviatilis* showed positivity for adrenocorticotropic hormone (ACTH) in *substantia nigra* (SN) cells and microglia [121]. Double-labeling experiments showed that a substantial proportion of microglial cells emigrating from cultured *substantia nigra* (SN) explants were adrenocorticotropic hormone (ACTH)-positive [121]. A few irregularly shaped ACTH-immunoreactive cells resembling microglia were also found in the neural lobe of *Diplodus sargus*, where microglia may exert macrophage-like effects [122]. ACTH and other peptides derived from the magnocellular nucleus of the preoptic area have broad and potent anti-inflammatory effects. Since activated microglial cells are an important source of cytokines and other inflammatory agents for the CNS, melanocortins may play an important role in controlling glial function to preserve neurons [123]. In this regard, the hypothesis that ACTH molecules are able to mediate communication between SN neurons and microglia in fish seems quite plausible.

Microglia are a heterogeneous population of CNS cells and play an active role in maintaining normal physiological conditions by scanning the cellular environment with ramified processes and undergoing rapid morphological changes in response to mediators such as ATP [117,124]. Various methods have been used to identify microglial cells in teleost fishes [117,119,125,126], of which lectin histochemistry has proven to be an effective tool for microglial detection. Lectins from *Lycopersicon esculentum* (LEL) have an affinity for poly-N-acetyllactosamine sugar residues and have provided the identification of amoeboid and branched microglial cells in the neonatal and adult rat brain, microglial cells in normal and injured retina, optic nerve and optic tube in fish, and a microglial cluster in the SN of *Tetraodon fluviatilis*, also in vitro [119,127,128]. During development, microglia dynamically interact with synapses, altering their structure and function, and are involved in the maintenance of brain homeostasis. Microglia interact with astrocytes to produce soluble factors that promote the integrity of the local neuronal environment. In addition, microglial-derived soluble factors contained in microglia appear to regulate the neurogenic differentiation of NSCs [119]. IL-6 and LIF molecules, released by activated microglia, promote the astrocytic differentiation of NSPCs, which indicates the importance of intercellular interactions between glial cells and NSPCs [129].

Ultrastructural studies of neuroglial relationships in the tegmental nuclei have revealed the presence of protoplasmic astrocytes that form tight contacts with large DTN neurons. In the tegmentum of juvenile chum salmon, the presence of astrocytes is associated with their involvement in maintaining the metabolism of large tegmental neurons and with the loss of neurogenic capacity due to astrogliogenesis. Microglial cells maintain the CNS in a quiescent state through adaptive plasticity. When microglia are injured, fish respond rapidly to a variety of signaling molecules, which indicates a disruption in the structural integrity of the brain, as well as to slight changes in the homeostasis of their microenvironment [130]. Studies on mammals are complemented by studies on teleost fish models, where microglia are active in clearing cellular debris [71,131], but also likely produce and respond to a variety of signaling molecules, in particular nitric oxide (NO) and cytokines, similar to those in mammals [132,133].

### 3.4. Neurogenic Zones of the Tegmentum in Juvenile Chum Salmon and Their Involvement in Adult Neurogenesis

An ultrastructural analysis of the neurogenic periventricular regions of the tegmentum in juvenile chum salmon showed that the RLTNZ, TSNZ, and LCTNZ contain cells of various types previously characterized in the telencephalon [1], as well as sensory niches located in the olfactory bulb, *tectum opticum*, and lobe X nerve in zebrafish [23]. In the RLTNZ of juvenile chum salmon, cells of types III and IVa were found to constitute the majority. In addition to these types, ependymal cells were observed in the dorsal ependymal membrane, and single elongated cells of the neuroepithelial type were also identified (Figure 7B,C).

The PVZs of the tegmentum, which is the suprasegmental motor center of the midbrain in fish, have not previously been studied. However, sufficient data have been accumulated on neurogenic zones in the sensory structures of zebrafish, such as the olfactory bulb, vagus, and facial lobes, as well as in the TeO [134,135,136].

The ultrastructural composition and regulation of these niches, the properties of neurogenic plasticity, and the feedback between the ability of sensory structures to process information and the phenotype of adult neural stem cells have been characterized in detail [23]. Studies on cichlids have shown that changes in the rate of cell proliferation in sensory and non-sensory niches can be associated with social status [28]. An experimental study of the relationship between the social environment and neurogenesis in adult zebrafish in the telencephalon and sensory niches has demonstrated that sensory niches are affected by social isolation or novelty, possibly due to differences in the level of sensory stimulation [23]. However, it remained unclear in this study whether sensory niches are modulated by multimodal or unimodal stimuli.

Studies of neurogenesis in the adult zebrafish forebrain have shown [50,137] that the cell types within the neurogenic niches consist of populations of proliferating and non-proliferating glia that can be classified using immunohistochemical markers, with the stem/progenitor cell phenotype typically having a RG profile [50,137]. TEM studies of the zebrafish forebrain have distinguished seven cell types that are present within the neurogenic niches of adult zebrafish [1].

According to the detailed characterization of the pallial and subpallial zones in zebrafish, pallial niches are dominated by superficially located RG-like type IIa cells, whereas subpallial niches are dominated by elongated type III cells reminiscent of NE-like phenotypes [1].

The detection of type III and IVa cells in the RLTNZ of juvenile chum salmon is consistent with the results of studies of the visual neurogenic niche (PGZ) of adult zebrafish [23]. According to Lindsey, type III cells contain elongated nuclei with heterochromatin and scanty cytoplasm and resemble neuroepithelial phenotypes [1]. However, we found a few cells that phenotypically resembled NECs with the respective ultrastructural features: elongated bodies, nuclei located in the basal part of the cell, an apical cilium, and a short basal process (Figure 7B). The detection of such cells in juvenile chum salmon is an interesting finding, which confirms fetalization—the retention of features of the embryonic structure—in the brain of 2-year-old chum salmon at the ultrastructural level. It was previously determined [3,13,22] that salmon fish are characterized by fetalization—the preservation of signs of the embryonic structure of the brain in adult animals. A characteristic feature of NECs is polarization along the apical–basal axis. A characteristic feature of the apical domain of NECs is the presence of a primary cilium [138].

NECs in the RLTNZ of juvenile chum salmon form adherens junctions at the apical end (Figure 7B,C). NECs are known to be characterized by two forms of mitosis: expansive symmetric divisions and differentiating asymmetric divisions [139]. In the RLTNZ of juvenile chum salmon, we found evidence for symmetric divisions of NECs (Figure 7C). As was previously shown for several cellular phenotypes in zebrafish, either the overexpression or downregulation of PAR complex proteins (par3, par6, and atypical protein kinase C) induces expansive symmetric divisions due to neurogenic asymmetric divisions [140].

A study of the adult zebrafish tectum identified a Her5-positive population of NECs located in the caudal part of the tectum [141]. This population of cells was in a state of slow cyclic proliferation, while retaining the Her5 labeling, which led to an increase in the number of progenitors lining the tectal marginal zone (TMZ). Although NECs in the TMZ were identified earlier [9], their origin remains unclear. A recent study of the medaka pallium has identified NEC-specific markers [20], which will help better characterize and compare NE stem/progenitor lineages in different neurogenic niches in adult fish. Other fish species such as zebrafish, masu salmon (*O. masou*) [64], and rainbow trout (*O. mykiss*) [16,142] also retain distinct NE and RG cells with NSC properties across in a range of neurogenic zones from embryonic development to adulthood [6,9,20,23]. The predominant NSC subtypes present in the mature CNS are the result of distinct developmental programs that may have different neurogenic and reparative potentials by adulthood [11]. The heterogeneity in the nature of NSCs is determined at multiple levels, including the molecular characteristics of NSCs, the cellular state (i.e., quiescent, slow cycling, and rapid gain transit), their glial or neurogenic identity, and, finally, the subtypes of glia and/or neurons they are capable of producing under physiological and pathophysiological conditions. The diversity of NSPC phenotypes across species, in addition to the heterogeneous nature of the stem cell niches themselves [137,143,144], highlight the need for a better understanding of the biology of these cells at the population level.

Most cells within the tegmental neurogenic zones of juvenile chum salmon at this age are type III and IVa cells, i.e., phenotypically equivalent to aNSPCs. In chum salmon, cells undergoing symmetric and asymmetric division were identified within the RLTNZ and LCTNZ (Figure 7A and Figure 9F). An IHC study of BLBP localization in the tegmental neurogenic zones of intact chum salmon and after acute traumatic injury showed the increased expression of this molecular marker, which is a marker of NSPCs in the zebrafish brain [11]. Studies of the adult zebrafish PGZ revealed a heterogeneous population of IIa, III, and IVa cells, of which some were labeled with BrdU and expressed glutamine synthetase and other glial markers [23]. The results of the study showed, in addition to BrdU, GS, and GFAP-expressing cells, BrdU, GS, and GFAP-negative type III and IVa cells in the neurogenic zones of the caudal tectum [23]. Based on these data, Lindsey et al. [23] suggested the existence of alternative phenotypes of aNSPCs in zebrafish that require further investigation. We also assume that the study of the tegmental neurogenic zones of juvenile chum salmon using TEM and IHC labeling will help identify the phenotypes of aNSPCs and determine their involvement in homeostatic and regenerative neurogenesis.

### 3.5. BLBP Expression in the Tegmentum of Intact Juvenile Chum Salmon and in the Post-Traumatic Period

The BLBP in intact juvenile chum salmon had a rather restricted expression pattern in the neurogenic zones (Figure 10A), but single immunopositive cells were present in the subventricular, parenchymal, and deeper zones of the tegmentum (Figure 10A,C,D), and the *valvula cerebelli* (Figure 10B). Intact animals also often showed dense clusters of negative undifferentiated cells forming centers of secondary neurogenesis (Figure 10E,F). In general, intact chum salmon showed low levels of BLBP expression in the tegmentum including, in particular, the neurogenic zones.

After acute injury, BLBP synthesis was significantly increased in cells and fibers of the RG, significantly intensified along the guides and vessels, and increased multifold in all areas of the tegmentum and *valvula cerebelli*. In the RLTNZ, a significant induction of the BLBP was also detected in cells and fibers of the RG (Figure 11A–D). After injury to the *medulla oblongata*, a multifold increase in BLBP immunopositivity was detected in the tegmentum of juvenile chum salmon in cells and fibers of the RG in the parenchyma and in the neurogenic zones in the peripheral areas (Figure 12A–F). The analysis of the distribution of BLBP+ cells in various areas of the tegmentum of intact juvenile chum salmon and after acute injury to the medulla oblongata showed a significant increase in their number in the acute post-traumatic period (Figure 18). As a result of the acute injury, the number of BLBP+ cells in the rostral tegmentum increased almost 30-fold, in the projection of the oculomotor nerve—30-fold, in the torus semicircularis—25-fold, and in the caudal tegmentum—almost 100-fold (Figure 18). This indicates the complex nature of acute post-traumatic changes in the motor and sensory tegmental areas when the abducent nerve projection (nVI) is damaged.

The results of mammalian cerebellum studies showed that the addition of affinity-purified anti-BLBP antibodies to cultured systems that analyze granule cell migration on glial substrates and the granule neuron induction of glial process extension indicates that the BLBP is involved in both neuronal differentiation and glial maintenance, migration, and differentiation [145].

The BLBP was previously shown to label RG cells in adult and larval zebrafish [26,47,48,49,50]. In the zebrafish tegmentum, BLBP immunolabeling was detected at the boundary of the TS, and numerous fibers were observed in the nucleus of the NFLM, extending along the vascular lacuna of the *area postrema*. BLBP staining was also found in the periphery of the *valvula cerebelli* and, to a lesser extent, in the cerebellar body. In all these regions in zebrafish, the BLBP is not co-expressed with the nuclear neuronal marker HuC/D, as was found in the subpallium [41]. This study also did not find any obvious differences in the distribution of the BLBP between the sexes.

Thus, in general, the major difference in BLBP distribution patterns between intact juvenile chum salmon and zebrafish was primarily that chum salmon does not have RG fibers in the tegmentum, and only a few intensely labeled cells without processes are found. In the *valvula cerebelli*, BLBP immunolocalization in chum salmon was mainly concentrated in the central part, not in the periphery, as in zebrafish. The functions of the BLBP are poorly understood to date, but it is generally recognized to play a key role in lipid accumulation, membrane synthesis, energy production through lipid transport into mitochondria, cholesterol metabolism, and cell signaling [47,50]. Its role in cell signaling is suggested to involve the interaction of fatty acids with transcription factors such as members of the peroxisome proliferator-activated receptor (PPAR) family [43]. The localization of BLBP mRNA and protein shows transient expression in Bergmann glial cells in the cerebellum during periods of granule cell migration along Bergmann fibers [146]. The data obtained through the examination of the developing cerebellar cortex are consistent with the localization of BLBP mRNA and protein in other parts of the developing mammalian CNS, where the BLBP is transiently expressed in RG cells and, possibly, in post-implantation neurons undergoing early differentiation [145]. These data suggest that the BLBP represents a novel neuronal signaling pathway that is important for CNS development.

The study of BLBP localization in the post-traumatic period in the tegmentum of juvenile chum salmon showed a multifold increase in its expression in a variety of systems. First, we identified complexes of immunopositive RG fibers in combination with immunopositive cells migrating along them (Figure 13A–D). Patterns of similar post-traumatic BLBP localization were numerous in all areas of the tegmentum, and BLBP overexpression was also detected in the RLTNZ.

The absence of RG fibers expressing the BLBP in intact chum salmon sharply contrasts with the data on zebrafish; however, the induction of BLBP expression in the RG after injury is consistent with the concept of glial differentiation. Mammalian cell culture experiments demonstrated the transient expression of the BLBP in glial cells that support neuronal migration. To clarify the functions of the BLBP in mammals and specific aspects of both glial and neuronal differentiation, in vitro studies were performed. A number of culture systems for purified cerebellar cell populations were developed that allow the functional analysis of cell–cell interactions at specific stages of granule cell development, including neuronal proliferation, neurite outgrowth, the neuronal induction of glial fiber extension, and neuronal migration along glial fibers [146,147]. The results of these in vitro experiments demonstrate that the BLBP is involved in glial differentiation by promoting the differentiation of radial processes that support neuronal migration, an important step in neuronal development. Based on these data, we suggested that in the post-traumatic period in juvenile chum salmon, the BLBP is involved in the RG differentiation, formation of its end apparatuses, and maintenance of the population of migrating BLBP+ cells during regeneration.

The local increase in BLBP expression in the post-traumatic period was also recorded from chum salmon DTN neurons (Figure 13D). These data are consistent with the results of immunoelectron microscopy in mammals, according to which the subcellular localization of the BLBP indicates the presence of the BLBP in both the cytoplasm and nucleus of granule neurons, suggesting a possible role of the BLBP in the neuron–glia signaling system [145]. We assumed that the increased BLBP expression in tegmental neurons in the acute post-traumatic period in chum salmon was also related with the involvement of the BLBP in post-traumatic neuronal differentiation. The role of the BLBP in neuronal differentiation is evidenced by the ability of anti-BLBP antibodies to block both neuronal and glial differentiation in mixed primary cerebellar cell cultures [146]. This system has developed in mammals [147,148] and has provided a large body of data indicating both homotypic and heterotypic cell–cell interactions in neuronal differentiation. In particular, the presence of large numbers of neurons surrounding primary cerebellar glia stimulate the differentiation of these glial cells into thin RG-like cells [149]. The latter, in turn, suppress neuronal proliferation and promote neuronal differentiation [148]. The addition of anti-BLBP antiserum to cerebellar cultures blocks both the extension of RG processes between granule cell precursor aggregates and the migration of granule cells along these processes, but does not affect either cell proliferation or adhesion, thus providing direct evidence that the BLBP is important for neuronal differentiation in vitro. Based on the results of these in vitro studies on the localization of the BLBP in the nucleus of Bergmann glial cells, and on the post-traumatic expression of BLBP in tegmental neurons of juvenile chum salmon, we assume that the BLBP is involved in the regulation of neuronal differentiation.

### 3.6. Aromatase Expression in the Tegmentum of Intact Juvenile Chum Salmon and in the Post-Traumatic Period

Aromatase, a key enzyme responsible for estrogen biosynthesis, is present in the brain of all vertebrates. There is ample evidence that in rainbow trout (*O. mykiss*), zebrafish (*D. rerio*), pejerrey (*Odontesthes bonariensis*), Japanese eel (*Anguilla japonica*), and medaka (*Oryzias latipes*), aromatase is highly expressed exclusively in proliferating mature RG cells of the brain even in adulthood, in contrast to other vertebrates [31,47,49,56,150,151]. However, the physiological significance of this expression remains unknown.

An analysis of AroB expression in different parts of the tegmentum in juvenile chum salmon showed a low content of AroB-immunopositive cells in the SVZ and parenchyma of intact animals (Figure 14A). A higher number of AroB-immunopositive cells were detected in the deep areas, at the level of the oculomotor nerve nuclei (Figure 14D) and the ventro-lateral zone (Figure 14F). No AroB-immunopositive RG fibers were detected in intact juvenile chum salmon.

In the post-traumatic period, AroB+ RG complexes with AroB+ cells migrating along vessels were detected in the ventro-lateral and lateral areas of the tegmentum, and large clusters of AroB+ cells were observed in the ventro-medial area (Figure 16A,B). Overall, the migration patterns of AroB+ cells were recorded from different regions of the tegmentum and from the *valvula cerebelli* (Figure 17B,C).

The analysis of the distribution of AroB+ cells in various areas of the tegmentum of intact juvenile chum salmon and after traumatic injury to the medulla oblongata showed a significant increase in their number in the acute post-traumatic period (Figure 19). As a result of the acute injury, the number of AroB+ cells in the isthmus increased 10-fold, in the rostral tegmentum—20-fold, and in the cerebellar flap—50-fold (Figure 19). This indicates the heterogeneous nature of post-traumatic changes in the tegmental area with damage to the abducent nerve projection (nVI). Combined damage to the cerebellum of juvenile chum salmon [36] also showed a significant increase in the number of AroB+ cells in various areas of the cerebellum. The main difference in damage to the medulla oblongata is the initial low density of the distribution of AroB+ cells in the tegmentum compared to the cerebellum, where the density of distribution of cells and the number of AroB+ cells in intact animals were slightly higher [36]. The results obtained on juvenile chum salmon are consistent with data on mammals showing that the aromatase enzyme is not expressed by glial cells under normal conditions, but under stress conditions such as serum starvation, which induces aromatase expression in cultured astrocytes [152]. In addition, acute brain injury induces aromatase expression in rodent astrocytes [57,152,153] and in avian RG [154,155,156]. In this context, increased aromatase expression in cells of the astroglial lineage was observed in all injured regions of the rodent brain, including the cerebral cortex, *corpus callosum*, striatum, hippocampus, thalamus, and hypothalamus [57]. Increased aromatase expression after brain injury was associated with increased aromatase activity [57] and increased brain estradiol levels [157].

Studies on zebra finches have shown that cyclooxygenase, prostaglandin E2, and neuroinflammation induce aromatase expression in glial cells [158,159]. In turn, the local synthesis of estradiol in the injured brain reduces neuroinflammation [158]. Indeed, aromatase expression by astrocytes and RG after brain injury may be part of the mechanisms supporting brain recovery after injury. Estrogens are considered “neuroprotective” in the adult brain because they exert protective effects by preventing excitotoxicity, inflammation, and oxidative damage; inhibiting apoptosis; promoting neuronal survival; regulating dendritic remodeling, synaptogenesis, and steroidogenesis; and modulating various neurotransmitters such as acetylcholine, dopamine, serotonin, glutamate, and GABA [152,160,161,162,163,164,165]. Studies on mammals have demonstrated increased neurodegeneration after injury induced by the central administration of aromatase inhibitors [152] or aromatase antisense oligonucleotides [166] and in an aromatase knockout compared to controls [152]. In addition, the peripheral administration of estradiol or aromatizable androgens reduced the lesion size in rats [167]. These results suggest an important role for glial aromatization in limiting the degeneration after lesion.

Overall, a study on the neuro-derivatives of estradiol, in particular aromatase, has shown their involvement in regulating the development and functional activity of various areas in the CNS that are not directly involved in the control of reproductive functions and behavior [38]. In addition, estradiol synthesized in the CNS has a neuroprotective effect after nervous system injury. Thus, we suggest that in juvenile chum salmon, the endogenous modulation of aromatase expression and activity in the context of CNS injury contributes to successful regeneration. Under intact and growing conditions, the local aromatase production in the CNS contributes to synaptic plasticity, behavioral modulation, and endogenous neuroprotection following neural injury.

Studies have also shown that variations in aromatase activity can influence androgen receptor and ERs signaling [168]. Since both testosterone and estradiol from the periphery can cross the blood–brain barrier, the final concentration of these steroids in the tegmentum depends on their content, local synthesis and metabolism, and their uptake from plasma. According to in vivo and in vitro studies, estradiol synthesis in the brain is necessary for the effects of peripheral estradiol on synaptic plasticity and neuroprotection [169,170]. On the other hand, recent studies indicate that the brain, as an endocrine organ, may promote circulating estrogens.

Mechanical or chemical injury has been shown to activate aromatase expression in reactive astrocytes surrounding the injury site or in RG cells facing the site, as observed in birds [57,154,171,172,173].

In our results from studies on the juvenile chum salmon cerebellum over a long period of time post-injury (90 days), the number of process-less type 1 cells increased in the cerebellum of juvenile *O. keta*, while the proportion of larger type 2 Aro+ cells (which may be analogous to mammalian reactive astrocytes) increased only slightly [36]. The results of the present study on chum salmon brainstem injury show a significant increase in type 2 cells in the ventrolateral and lateral regions of the tegmentum along the major bundles and vessels (Figure 16 and Figure 17). Thus, our results support that aromatase expression in chum salmon astroglial cells may be part of the mechanisms supporting brain recovery after injury [171,172,173,174].

Studies on *D. rerio* using the proliferation markers BrdU, proliferating cell nuclear antigen (PCNA), and aromatase B have shown that Aro+ RG cells actively divide to form new cells [31]. These newborn cells can undergo further division, migrate along the radial processes, and eventually differentiate into neurons [26,31,50].

## 4. Materials and Methods

### 4.1. Experimental Animals

This study was conducted on 25 Pacific chum salmon (*O. keta*) aged 2 years and 1 month, with a body length of 23.5–27 cm and a weight of 79–95 g. The animals were provided by the Ryazanovka Experimental Fish Hatchery in 2022. Most of the fish used in this study were males. Juvenile chum salmon were kept in a tank with aerated fresh water at a temperature of 13–14 °C and fed once a day. The day/night cycle was 14/10 h. The concentration of dissolved oxygen in the water was 7–10 mg/dm^3^, which corresponds to normal saturation. The study was carried out by the A.V. Zhirmunsky National Scientific Center of Marine Biology, Far Eastern Branch, Russian Academy of Sciences (NSCMB FEB RAS) and the Ethics Committee governed the humane treatment of experimental animals (approval No. 1-231024 dated 23 October 2024 at the meeting No. 10 of the Biomedical Ethics Committee of the NSCMB FEB RAS). The animals were divided into three groups. The animals of the control group (group 1) were intact (*n* = 10); the experimental groups consisted of fish subjected to acute traumatic injury of the medulla oblongata (group 2, *n* = 10) and intact animals for ultramicroscopic examination (TEM) (group 3, *n* = 5).

### 4.2. Transmission Electron Microscopy

Transmission electron microscopy was used to investigate the ultrastructural profile of neurons and glia within the nuclei of the mesencephalic tegmentum and within the neurogenic niches located in the periventricular zone of the tegmentum and the caudal TeO of the PGZ. These regions were selected because in juvenile chum salmon, they represent the dominant neurogenesis centers of the *torus semicircularis* [23] and the sensory structures (TeO) involved to process visual information.

After fixation of the mesencephalon in 2.5% glutaraldehyde in 0.2 M cacodylate buffer (pH 7.4) overnight at 4 °C, the samples were washed and then post-fixed for 2 h with 1% OsO_4_ in 0.2 M cacodylate buffer. The tissues were then washed and dehydrated in an ascending ethanol series and then impregnated with 100% ethanol and LR-White resin. The following day, the tissues were impregnated with fresh LR-White resin twice over a 6 h period and then embedded in gelatin capsules and polymerized at 40 °C.

Using a Leica UC7 ultramicrotome (Leica, Wetzlar, Germany,), semi-thin (500 nm) sections were cut onto glass slides and ultrathin (50–60 nm) sections were cut onto formvar-coated copper grids. The sections were made through the tegmentum and tectum at the appropriate rostro-caudal level, taking the PVZ into account. Semi-thin sections were stained with 1% methylene blue in 1% aqueous sodium tetraborate. Ultrathin sections were stained with 2% aqueous uranyl acetate and lead citrate. Visualization was performed using a Zeiss Sigma 300 VP transmission electron microscope (Carl Zeiss, Cambridge, UK) and the AMT Image Capture Engine software (version 5.44.599).

For the analysis of the morphology of cells present in the PGZ and isthmus niches, they were compared with the seven previously published ultrastructural profiles (Types IIa, Type IIb, Type III, Type IVa, Type IVb, Type V, and Type VI) previously described in [1,23].

### 4.3. Cell Counting and Visualization

Visualization was performed using an AxioImager Z2 microscope (Carl Zeiss, Gottingen, Germany). BLBP+ cells were quantified by counting at least every other section along the rostro-caudal neuraxial axis of each niche at 400× magnification for each biological sample (Figure 6B,C and Figure 7B,C; entire red area of each niche).

To examine the ultrastructural organization of the PVZ tegmentum and tectum, low-power (2–4K) TEM images were taken at rostro-caudal levels. These images were used to delineate the boundaries of each PVZ for the subsequent analysis of cell types and cell frequencies between niches. For the morphological analysis of the different cell types within the tegmentum nuclei that make up the PVZ, at least three cells of each type with morphologically unique ultrastructural features were selected from each PVZ and throughout the mesencephalic tegmentum and tectum. When only a few cells of a given morphology could be observed within a PVZ, all identified cells were analyzed.

To characterize the cell types, the following features were examined: outline, color, chromatin organization, number of nucleoli, length of the long and short axes of the nucleus; percentage, color, presence of mitochondria, cilia, microvilli, vacuoles, lipid droplets, dense bodies in the cytoplasm; localization of cell types; and cell–cell contacts. Using these criteria, the morphological profile and frequency of each cell type within each PVZ and as a percentage of all cells examined were calculated. Based on the identified morphological features, we created a model representing the cellular organization of the PVZ of the tegmental region and a classification scheme for the different cell types that make up different niches.

### 4.4. Experimental Design: Acute Traumatic Injury to Medulla Oblongata

To study the characteristics of the cellular response as a result of acute traumatic injury (group 2), animals were subjected to mechanical injury of the *medulla oblongata* in the area of the projection of the abducens (VI) nerve, followed by observation for 3 days after the injury. The injury of the *medulla oblongata* was caused by puncturing the fish skull with a thin sterile needle 3 mm deep in the parasagittal direction. The injury zone covered the area of the projection of the VI nerve in the *medulla oblongata*, located directly behind the cerebellum and did not affect other parts of the brain. Immediately post-injury, the animals were released into the aquarium for their recovery and further monitoring. On day 3 post-injury, the animals (group 2) were removed from the experiment and subjected to immunohistochemical examination.

### 4.5. Preparation of Material for Immunohistochemical Studies

*Anesthesia, prefixation after acute injury*. Fish were anesthetized with 0.1% tricaine methanesulfonate (MS222) (Sigma, St. Louis, MO, USA, Cat.# WXBC9102V) for 10–15 min. After anesthesia, the intracranial cavity of the immobilized animal was perfused with 4% paraformaldehyde (PFA, BioChemica, Cambridge, MA, USA; Cat.# A3813.1000; Lot 31000997) in 0.1 M phosphate buffer (Tocris Bioscience, Minneapolis, MN, USA; Cat.# 5564, Lot# 5, pH 7.4). Animals were sacrificed and euthanized by rapid decapitation. After preliminary fixation, the brain was removed from the cranial cavity and fixed for 24 h in 4% paraformaldehyde in 0.1 M phosphate buffer. Then, they were kept in 30% sucrose solution at 4 °C for two days (with seven changes of the solution). Serial frontal (50 μm thick) sections of the brain were cut on a freezing microtome (Cryo-star HM 560 MV, Waldorf, Germany). Every third frontal section of the mesencephalon was taken for reaction.

### 4.6. Immunohistochemical Verification of AroB and BLBP

The expression of the BLBP and AroB (cyp19a1b) markers was studied in intact juvenile chum salmon at 3 days after traumatic injury to the *medulla oblongata*. Before IHC, endogenous peroxidase activity and non-specific staining (background) were blocked by incubation with 1% hydrogen peroxide for 20 min at room temperature. To eliminate non-specific staining, brain sections were incubated with non-immune horse serum. Immunoperoxidase labeling was performed using monoclonal mouse antibodies against the BLBP and AroB at a dilution of 1:300 on frozen free-floating brain sections at 4 °C for 48 h (Table 5).

After incubation and washing with 0.1 M PBS, frozen brain sections were incubated in situ with primary goat anti-rabbit antigen-cyp19a1b (Abcam, Cambridge, UK; catalog number ab106168) and rabbit polyclonal antibodies against the BLBP (Abcam, Cambridge, UK; catalog number ab32423), diluted 1:300 at 4 °C for 48 h. The anti-rabbit streptavidin–biotin imaging system (HRP conjugated Anti-Rabbit IgG SABC Kit; Boster Biological Technology, Pleasanton, CA, USA; Catalog No. SA1022) was used to visualize the immunohistochemical labeling of mouse monoclonal antibodies, respectively. To identify the reaction products, a red substrate (VIP Substrate Kit, Cat. No. SK-4600, Vector Laboratories, Burlingame, CA, USA) was used according to the manufacturer’s recommendations.

Mesencephalon sections were placed on polylysine-coated glass slides (BioVitrum, St. Petersburg, Russia) and left to dry completely. To detect immunonegative cells, mesencephalon sections were additionally stained with 0.1% methylene blue (Bioenno Lifescience, Santa Ana, CA, USA, Cat. No. 003027). Stain development was observed under a microscope, washed in three changes of distilled water for 10 s, then differentiated for 1–2 min in a 70% alcohol solution and then for 10 s in 96% ethanol. The sections were dehydrated using the standard procedure: two changes of xylene for 15 min. Then, they were placed under coverslips in a Bio-optica medium (Milan, Italy). To assess the specificity of the immunohistochemical reaction, a negative control method was used (Figure 20). Instead of primary antibodies, mesencephalon sections were incubated with a 1% solution of non-immune horse serum for 1 day and processed in the same way as sections with primary antibodies. In all control experiments, no immunopositive reaction was observed.

### 4.7. Microscopy

A research-grade motorized inverted microscope with an attachment for improved contrasting Axiovert 200 m luminescence with an ApoTome fluorescence module and AxioCam MRM and AxioCam HRC digital cameras (Carl Zeiss, Jena, Germany) was used for the visualization and morphological and morphometric analysis of cell body parameters (measurements of the greater and lesser soma diameters). The material was analyzed using AxioVision (Axiovert 200 M, Axiovision software version 4.8; Carl Zeiss, Jena, Germany). Measurements were made at 100×, 200×, and 400× magnification and in several randomly selected fields of view for each region of interest. The number of labeled cells per field of view was counted at 200× magnification. Micrographs of the mounts were taken with an Axiovert 200 digital camera. The material was processed using the Axioimager program and the Corel Photo-Paint 17 graphic editor.

### 4.8. Densitometry

Optical densities (OD) of IHC-labeled products in neuronal cell bodies and immunopositive granules were measured using an Axiovert 200 M microscope (Axiovert 200 M, Axiovision software version 4.8; Carl Zeiss, Jena, Germany). The Wizard program was used to perform a standard assessment of the optical density for 5–7 sections by selecting 10–15 intensely/moderately labeled and immunonegative cells of the same type for analysis. The average OD for each cell type was then subtracted from the maximum OD for immunonegative cells (background), and, thus, the actual value in relative units of optical density (UOD) was obtained.

### 4.9. Statistical Analysis

Morphometric data of IHC labeling were quantitatively processed using the Statistica version 12 and Microsoft Excel 2010 and STATA software packages (version 12, tataCorp LP., College Station, TX, USA). All data were presented as the mean ± standard deviation (M ± SD) and analyzed using the SPSS software application (version 12.0; SPSS Inc., Chicago, IL, USA). All group changes were compared using the Student–Newman–Keuls test with the Bonferroni correction. Values at *p* ≤ 0.01 and *p* ≤ 0.001 were considered statistically significant.

## 5. Conclusions

Thus, as a result of the ultrastructural study of the tegmental nuclei in juvenile chum salmon, we identified neurons of three size types with a high metabolic rate, characterized by large numbers of mitochondria, polyribosomes, Golgi apparatus, and cytoplasmic inclusions (vacuoles, lipid droplets, and dense bodies). We also identified large synaptic terminals, presumably corresponding to acetylcholinergic synapses, in the NFLM. No similar structures were identified in other tegmentum nuclei and are, thus, an interesting finding for the NFLM, whose neurons originate in a multicomponent descending motor pathway in the mesencephalic tegmentum of 2-year-old juvenile chum salmon. However, in addition to synaptic neurotransmission, paracrine neurotransmission continues to exist in the tegmental nuclei, which excretes vesicles containing specific syntheses into the intercellular space. We suggest that the paracrine type of neurotransmission is characteristic of earlier developmental stages in juvenile chum salmon, before maturation and at the onset of synaptic connections.

In the suprasegmental nuclei of the tegmentum NFLM and DTN in intact juvenile chum salmon, cases of apoptosis of medium- and large-sized neurons of an unknown nature were identified. We suggested that the elimination of neurons by apoptosis in the tegmentum may be caused by various factors, including intrinsic ones, related to damage to cells and/or their genomes and, ultimately, by the activation of various death receptors that are expressed on the surface of most cells. Other causes of the apoptosis developing in tegmental neurons may be related to changes in the external environment that trigger programmed cell death.

Brainstem injury responses in chum salmon activate multiple signaling pathways that significantly increase the BLBP and AroB expression in different regions of the tegmentum and *valvula cerebelli*. However, post-traumatic patterns of the BLBP and AroB expression are not consistent. In addition to a general increase in BLBP expression in the tegmental parenchyma, the BLBP is overexpressed in the RLTNZ, whereas AroB expression is completely absent from the RLTNZ. Another difference is the peripheral overexpression of AroB and the formation of dense reactive clusters in the ventro-medial tegmentum.

During the post-traumatic response, there is a significant change in energy metabolism, in which astrocytes play an important role [175]. Special attention should be paid to the detection of specialized astrocytic processes in the tegmentum of juvenile chum, known as “astrocytic legs”, which were identified by the TEM method, as well as during the IHC labeling of the BLBP in traumatic conditions. This confirms the data previously established by the TEM method in mammals, which show that the “astrocytic legs” establish direct contact with vessels and capillaries, attaching them to the vascular surface [176]. According to our data, damage to the medulla oblongata of juvenile chum salmon leads to an increase in total and capillary blood flow in the anterior and caudal vascular plexuses. Changes in the diameters of blood vessels, especially at the capillary level, entail a change in blood flow in the local vascular network. In turn, these changes affect the distribution of vital substances necessary to maintain neural metabolism, including lactate, glucose, and oxygen [177].

The astrocytic response in the tegmentum of juvenile chum salmon occurs due to traumatic damage, with damage to neurons and the release of glutamate and other neurotransmitters that initiate astrocyte activation. This response is transmitted to neighboring astrocytes, causing changes in the perfusion of nearby capillaries. These changes in perfusion may be the result of either vasodilation or vasoconstriction in capillaries [178,179]. Thus, in the post-traumatic period, various pathways are activated, whose components are excellent candidates for the role of inducers of an “astrocyte-like” response in the brain of juvenile *O. keta*, similar to those present in the mammalian brain. In this case, the BLBP acts as a factor enhancing both RG and neuronal differentiation. Estradiol from Aro+ astrocytes exerts a paracrine neuroprotective effect through the potential inhibition of inflammatory processes. Our results indicate a new role of neuronal aromatization as a mechanism to prevent the development of neuroinflammation. Furthermore, these results confirm the hypothesis that the BLBP is a factor that promotes neuronal and glial differentiation in the post-traumatic period in the *O. keta* brain.

## 6. Limitations and Prospects

The conducted studies are the basis for the further study of the ultrastructural organization and IHC profile of the chum tegmentum to characterize neuronal plasticity and regenerative properties of the central nervous system. In this sense, the salmonid fish model is similar to mammals; in particular, astroglia have been identified in the brain of juvenile salmon, protoplasmic astrocytes have been identified, and RG is represented to a lesser extent. This is a significant advantage compared to other model objects, since no astrocytes have been detected in the brain of zebrafish, but RG cells are present. Thus, despite the fact that the study discusses zebrafish and other models, the extrapolation of the results to other species, including mammals, remains limited.

The possible applied consequences of our results imply their use, first of all, as potential strategies for activating aNSCPs in human CNS injuries. The numerous cell groups with immunopositivity to AroB and BLBP identified in acute traumatic injury in juvenile chum salmon make it possible to better understand how and which cellular systems in juvenile chum salmon are involved in the regenerative response, which is quite successful compared to mammals. Following this logic, it can be expected that the impact, including pharmacological effects on individual components of the cellular cascade involved in the post-traumatic regeneration of the human brain, may contribute to a more successful regenerative response. For example, increased aromatase B activity in astrocytes naturally leads to an increase in estradiol levels in the post-traumatic period, exerting a neuroprotective effect through the potential inhibition of inflammatory processes. The BLBP+ terminal apparatuses of astrocytes and/or pericytes identified in the post-traumatic period, and in some cases RG, contribute to changes in the vascular lumen, affecting the level of hemodynamics. This study is a pilot and implies the further study of ultrastructural and IHC processes in the brain of juvenile chum salmon, which has marked differences from other models for studying neurogenesis.

Another limitation of the present study is the small sample size for morphometric and densitometric analyses, which is not detailed, which potentially limits the statistical power and the ability to generalize the results. The data presented in this study from the statistical analysis of AroB+ and BLBP+ cell groups in the tegmental area of the brain of juvenile chum salmon reflect a pronounced tendency to an increase in the number of cells, which corresponds to data from damage to the cerebellum of chum salmon [36]. We believe that the extensive increase in the number of AroB+ and BLBP+ cells in case of injury to the medulla oblongata (projection of the nucleus of the VI nerve) of juvenile chum salmon is based on a pronounced anti-inflammatory response that occurs in response to injury in the acute period.

Functional analysis data, in particular, behavioral assessment, confirm the effect of the observed cellular changes (Appendix A). The results of the video monitoring of the mobility and behavioral reactions of juvenile chum salmon show that in intact conditions, they are characterized by a high level of mobility and a light body color adapted to light conditions (Appendix A). Immediately after traumatic injury, the mobility of animals decreases and the darkening of the dorsal part of the body is observed (Appendix A). However, 3 days after injury, the motor activity of juvenile chum salmon is restored, while the dark coloration of the dorsal part of the body remains. Since this study is a pilot study, the present work investigated a limited 3-day post-traumatic period, and in the future, we plan to expand post-traumatic monitoring to expand the understanding of the dynamics of neurogenesis and glial cell response over time.

## Figures and Tables

**Figure 1 ijms-26-00644-f001:**
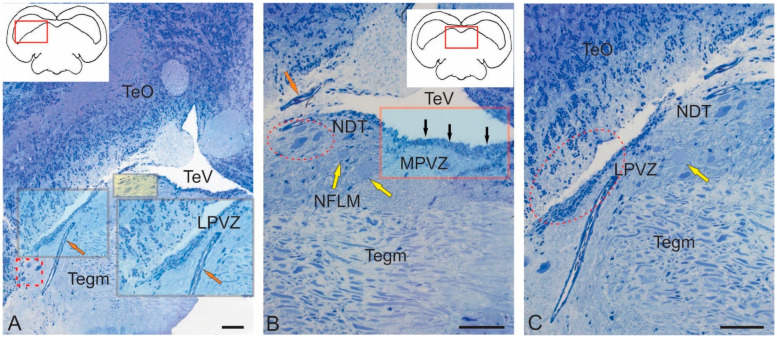
Methylene blue-stained semi-thin sections showing the major nuclear structures of the tegmentum and the periventricular region in juvenile chum salmon, *Oncorhynchus keta*. (**A**) The tegmentum region containing the lateral periventricular zone (LPVZ) is shown in the red box and inset; the orange arrow indicates the vessel; lateral tegmental neurons are outlined by a red dotted square; dorsal tegmental neurons (DTN) are shown in the yellow box; the pictogram in the red box shows the area of interest. TeO—optic tectum; Tegm—tegmentum. (**B**) The medial tegmentum region (is shown in the brown box) containing the medial periventricular zone (MPVZ); black arrows in the red box probably indicate aNSPC; yellow arrows indicate neurons of the nucleus of the longitudinal medial fasciculus (NFLM). (**C**) An enlarged fragment of LPVZ (in red dotted oval). Scale: (**A**–**C**) 100 µm.

**Figure 2 ijms-26-00644-f002:**
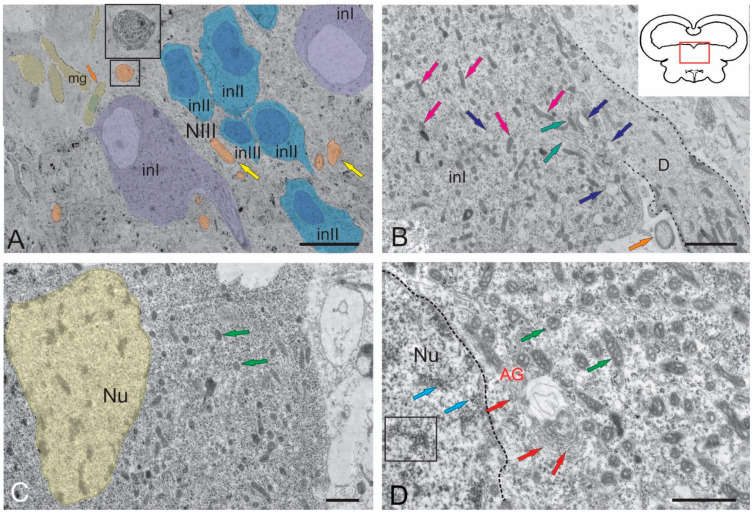
Ultrastructural organization of nucleus of oculomotor nerve III in the juvenile chum salmon, *Oncorhynchus keta*. (**A**) Large interneurons (inI) are shown in light purple; smaller interneurons (inII) are highlighted in blue; nuclei of inI and inII are highlighted in lighter and darker shades, respectively; interneuron with the smallest soma size is designated as inIII; myelin fiber bundles are highlighted in orange; the inset in the black square shows a cross section of a myelin fiber; the microglia population is highlighted in yellow; a microgliocyte with a nucleus is indicated by the orange arrow, myelin fibers are shown by the yellow arrow. (**B**) Ultrastructural organization of inI; pictogram shows the area of NIII localization; the dotted line indicates dendrite (**D**); pink arrows indicate mitochondria; blue arrows indicate lipid droplets; green arrows indicate actin microfibrils; the orange arrow indicates a myelin fiber. (**C**) Ultrastructural organization of inI at higher magnification; the nucleus is highlighted in yellow; mitochondria are indicated by green arrows. (**D**) Ultrastructural organization of inI at higher magnification; nuclear boundaries (Nu) are indicated by the black dotted line; the heterochromatin fragment containing polyribosomes is outlined by the black rectangle; single polysomes are indicated by blue arrows; Golgi apparatus cisterns (AG) are indicated by red arrows; mitochondria are indicated by green arrows. Transmission electron microscopy. Scale: (**A**) 10 μm; (**B**) 2 μm; and (**C**,**D**) 1 μm.

**Figure 3 ijms-26-00644-f003:**
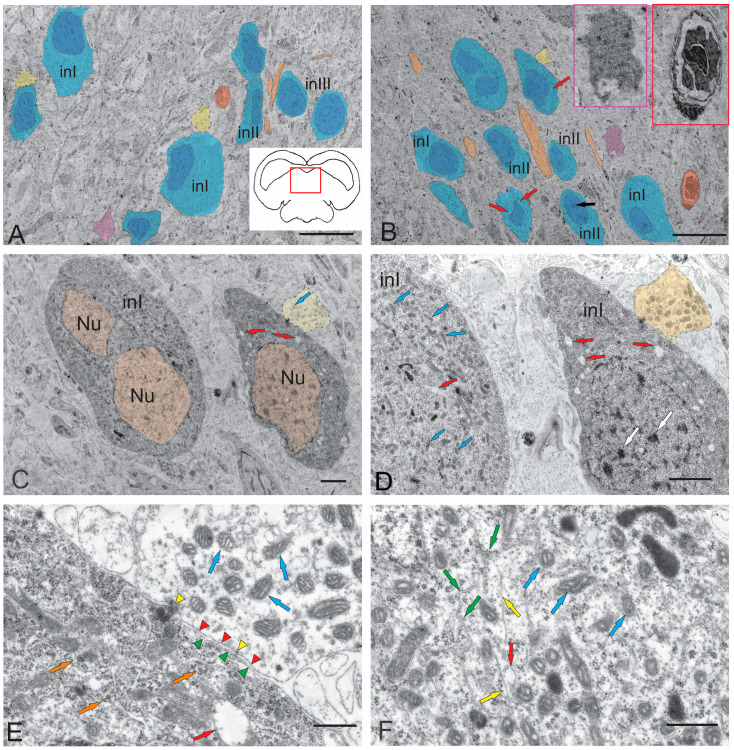
Ultrastructural organization of nucleus of *fasciculus longitudinalis medialis* (NFLM) in juvenile chum salmon, *Oncorhynchus keta*. (**A**) Ipsilateral (right) part of NFLM, large inI, medium inII, and small inIII interneurons are highlighted in light blue; nuclei, in blue; synaptic terminals, in yellow; oligodendrocyte, in light purple; apoptotic cell, in light red; fibers, in orange. (**B**) Contralateral part of NFLM; lipid droplets are indicated by red arrows; nucleolus, by a black arrow; enlarged oligodendrocyte, in the purple rectangle; enlarged apoptotic cell, in the red rectangle; other designations are as in (**A**). (**C**) Enlarged fragment containing large type I interneurons in B; inI nuclei are highlighted in light orange; mitochondria in the synaptic terminal are indicated by light blue arrows; other designations are as in (**B**). (**D**) Ultrastructural organization of inI at higher magnification; nucleoli are indicated by white arrows; other designations are as in (**C**). (**E**) Ultrastructural organization of asymmetric synaptic terminal; tight junction zone is indicated by yellow arrowheads; presynaptic membrane, by red arrowheads; postsynaptic membrane, by green arrowheads; polyribosomes, by orange arrows; other designations are as in (**D**). (**F**) Ultrastructural organization of the presynaptic region at a higher magnification; neurofilaments are indicated by yellow arrows; rough endoplasmic reticulum cisterns with polyribosomes, by green arrows; other designations are as in (**E**). Transmission electron microscopy. Scale: (**A**,**B**) 10 μm; (**C**,**D**) 2 μm; and (**E**,**F**) 500 nm.

**Figure 4 ijms-26-00644-f004:**
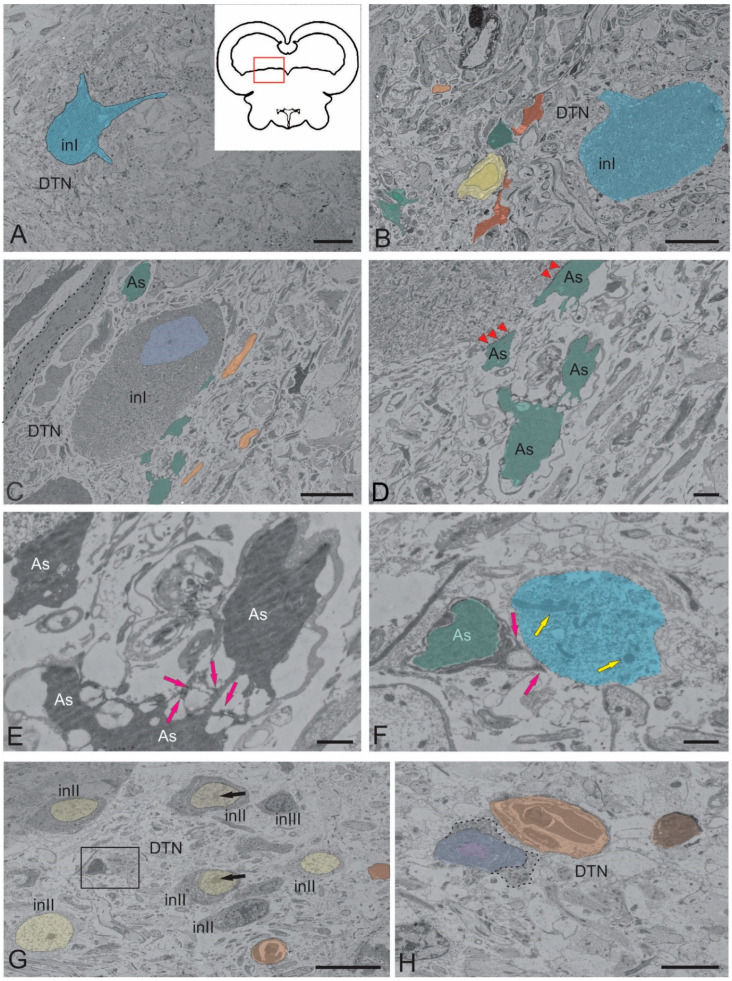
Ultrastructural organization of dorsal tegmental nucleus (DTN) in juvenile chum salmon, *Oncorhynchus keta*. (**A**) Large neuron with primary dendrites of DTN is highlighted in blue. (**B**) Large interneuron with lipid inclusions is highlighted in blue in parenchyma of the dorsomedial part of tegmentum; oligodendrocytes are highlighted in light red; astrocyte, in green; medium-sized interneuron inII is highlighted in yellow; and cross section of myelinated fiber, in orange. (**C**) Large primary interneuron (nucleus is highlighted in light blue) in contact with protoplasmic astrocytes (in green); a fragment of longitudinal myelinated fiber is outlined by a dotted line; other fragments are highlighted in orange. (**D**) An enlarged fragment containing neuroglial somato-somatic tight junctions (red triangular arrows) of a large interneuron and astrocytes (As, highlighted in green). (**E**) Protoplasmic type astrocytes (As) with short, branched processes ending with characteristic “astrocytic legs” (magenta arrows). (**F**) A fragment of a neuron (in blue) in contact with the “astrocytic legs” (indicated by magenta arrows) of a protoplasmic astrocyte (in green); mitochondria are indicated by yellow arrows. (**G**) Interneurons of second type inII (nuclei are highlighted in yellow) and third type inIII; nucleoli are indicated by a black arrow; in DTN, neurons in a state of apoptosis and apoptotic body are highlighted in orange; an enlarged fragment is shown in the black rectangle in (**G**). (**H**) Patterns of apoptosis of large neuron and apoptotic body (highlighted in orange) and paracrine neurosecretion (neurosecretory cell is indicated in light purple); neurosecretory granules are outlined by a dotted line. Transmission electron microscopy. Scale: (**A**–**C**,**G**) 10 µm; (**D**) 2 µm; (**E**,**F**) 1 µm; and (**H**) 5 µm.

**Figure 5 ijms-26-00644-f005:**
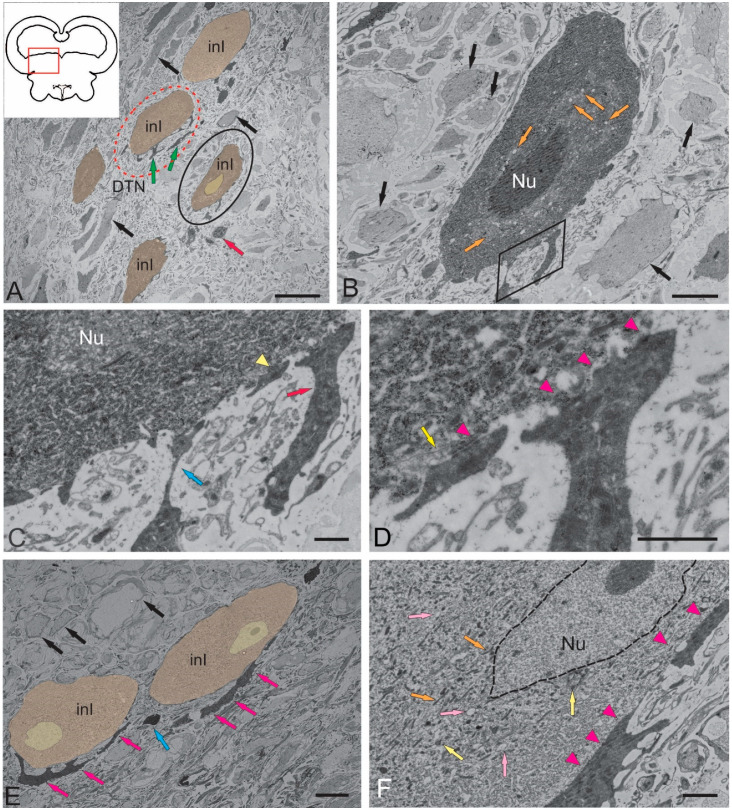
Ultrastructural features of neuro–glial relationships in DTN of juvenile chum salmon, *Oncorhynchus keta*. (**A**) Large inI interneurons (cytoplasm is highlighted in yellow, in red dotted oval) in contact with protoplasmic astroglia (indicated by green arrows); myelinated fibers are indicated by black arrows; oligodendrocyte, by a magenta arrow. (**B**) Ultrastructural organization of inI interneuron shown in (**A**) (in a black oval) at a higher magnification; lipid droplets are indicated by orange arrows; Nu—nucleus; fragments of myelinated fibers are indicated by black arrows. (**C**) Enlarged fragment shown in (**B**) (outlined by black diamond); the “astrocytic legs” are indicated by a blue arrow; yellow arrowhead indicates the zone of tight contact between the astrocyte and inI; red arrow indicates the area of extended contact. (**D**) Enlarged fragment in (**C**) showing the area of desmosome-like structures (magenta triangular arrows) in the zone of tight neuroglial junctions and neurofilaments (indicated by yellow arrow). (**E**) DTN neurons of deeper parenchymatous localization (cytoplasm highlighted in light orange; nuclei, in yellow) contacting astrocytic glia (indicated by magenta arrows); a fragment of astrocyte is indicated by a blue arrow; myelin fibers, by black arrows. (**F**) An enlarged fragment in E showing neuro–glial contacts (magenta triangle arrows); small mitochondria are indicated by yellow arrows; elongated mitochondria, by orange arrows; lipid droplets, by pink arrows. Transmission electron microscopy. Scale: (**A**) 20 μm; (**B**) 5 μm; (**C**,**D**) 1 μm; (**E**) 10 μm; and (**F**) 2 μm.

**Figure 6 ijms-26-00644-f006:**
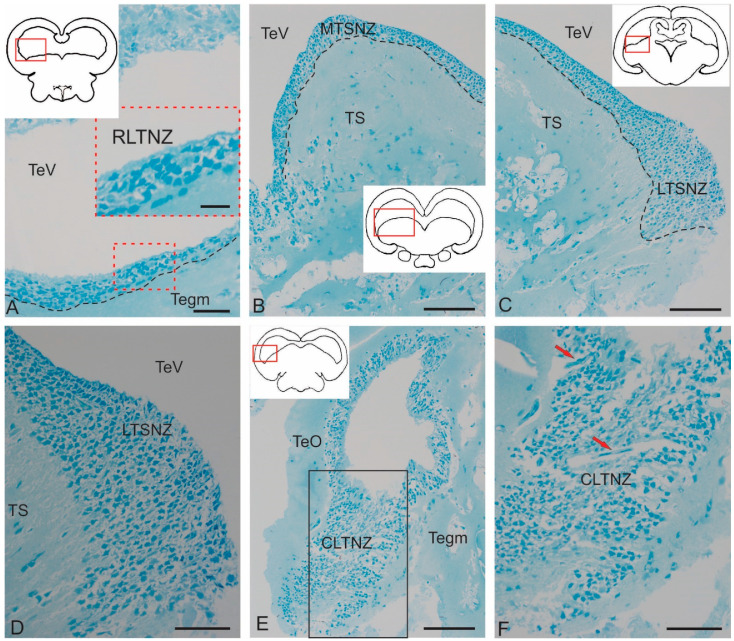
Semi-thin sections stained with methylene blue showing the tegmental and tectal neurogenic zones in the latero-caudal region of *torus semicircularis* in juvenile chum salmon, *Oncorhynchus keta*. The red rectangle on the pictograms shows the brain regions in the micrographs. (**A**) Rostral lateral tegmental neurogenic zone (RLTNZ); the basal border is indicated by the black dotted line; TeV—tectal ventricle; an enlarged fragment is in the red dotted inset. (**B**) Neurogenic zone of *torus semicircularis* (TS), medial part (MTSNZ). (**C**) Lateral neurogenic zone of *torus semicircularis* (LTSNZ). (**D**) Enlarged fragment of LTSNZ. (**E**) Caudal segment of the lateral tegmental zone CLTNZ (in black rectangle). (**F**) Enlarged fragment of CLTNZ; vessels are indicated by red arrows. Scale: (**A**,**D**,**F**) 50 µm and (**B**,**C**,**E**) 100 µm.

**Figure 7 ijms-26-00644-f007:**
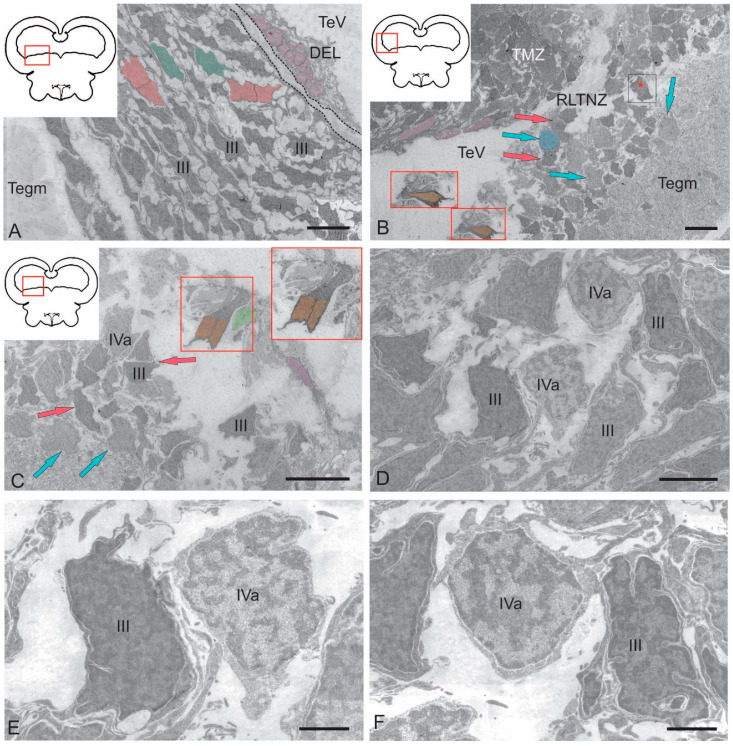
Ultrastructural organization of the rostral lateral tegmental neurogenic zone (RLTNZ) in juvenile chum salmon, *Oncorhynchus keta*. (**A**) Lateral part of RLTNZ (shown in the red rectangle in the pictogram); dorsal ependymal lamina (DEL) consisting of a pseudo-monolayer of cuboidal ependymal cells (highlighted in pink); type III cells identified in RLTNZ (highlighted in green); divided cells are highlighted in pink. (**B**) RLTNZ of tegmentum (Tegm) at the border with tectal marginal zone TMZ; elongated ependymal cells are highlighted in pink; type III cells are indicated by red arrows; type IV cells, by blue arrows; neuroepithelium-like cell is in the red rectangle; cytoplasm is highlighted in orange; apical contact zone is indicated by the asterisk. (**C**) Medial RLTNZ containing type III and IVa cells, ependymal cells, and neuroepithelial-like cells in mitosis; apical NEC contacts superficial layer aNSPCs (in green), as in (**B**). (**D**) Deep RLTNZ containing aNSPCs (type III and IVa cells). (**E**) Morphological heterogeneity of type III and IVa cells from medial RLTNZ at a higher magnification. (**F**) Light nucleus and heterochromatin in type IVa cells and highly irregular dark nucleus of type III cells from lateral RLTNZ at a higher magnification. Transmission electron microscopy. Scale: (**A**,**D**) 5 μm; (**B**,**C**) 10 μm; and (**E**,**F**) 2 μm.

**Figure 8 ijms-26-00644-f008:**
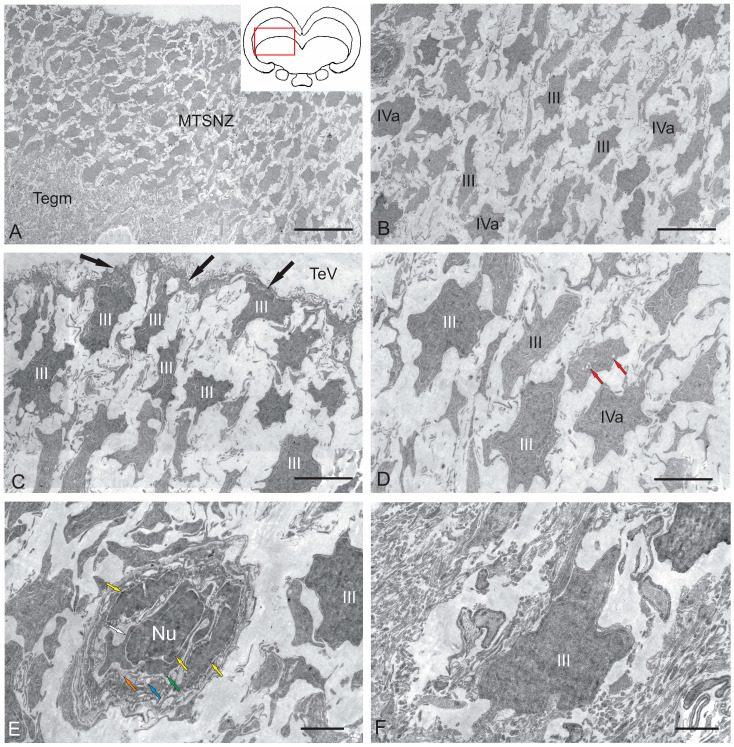
Ultrastructural organization of the neurogenic zone of *torus semicircularis* (TSNZ, red box in pictogram) in juvenile chum salmon, *Oncorhynchus keta*. (**A**) aNSPCs in the medial part of the neurogenic zone of *torus semicircularis* (MTSNZ); pictogram shows the area of study. (**B**) Cells of types III and IVa in the ventral part of MTSNZ at a higher magnification. (**C**) Cells of type III in the dorsal part of MTSNZ contacting apical membrane (indicated by black arrow). (**D**) Cells of types III and IVa in TSNZ with large nuclei of irregular shape and minimal cytoplasm containing lipid droplets (red arrows). (**E**) Microvessel in TSNZ with endothelial cells (yellow arrows) localized in its wall; in the central part, an immature form of a macrophage (white arrow) containing a nucleus (Nu) and branched cytoplasmic processes (green arrow), lipid inclusions (blue arrow), and endothelial cell cytoplasm (orange arrow). (**F**) Type III cells in TSNZ at a higher magnification, in contact with the intercellular substance. Transmission electron microscopy. Scale: (**A**) 20 μm; (**B**) 10 μm; (**C**,**D**) 5 μm; and (**E**,**F**) 2 μm.

**Figure 9 ijms-26-00644-f009:**
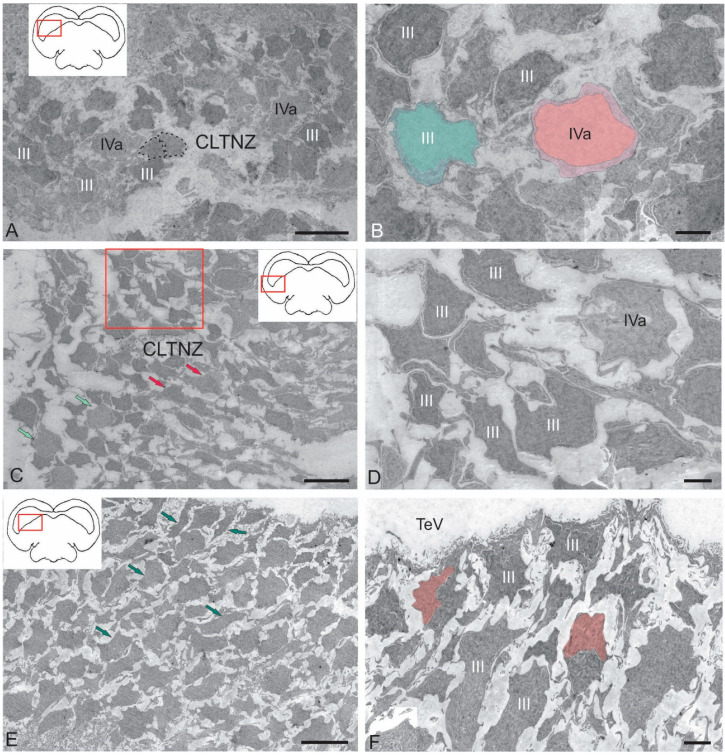
Ultrastructural organization of latero-caudal tegmental neurogenic zone (CLTNZ). (**A**) Cellular structure of CLTNZ (pictogram shows the area of study); cells of types III and Iva; dividing cells are indicated by the dotted line. (**B**) Morphological phenotypes of cells of types III (highlighted in green) and IVa (in red); cytoplasm of cells is highlighted in lighter tones; the nucleus is highlighted in a more saturated tone. (**C**) Caudal part of CLTNZ containing type III cells with nuclei having irregular and elongated morphologies (indicated by red arrows) and more ventrally localized cells of type IVa (indicated by green arrows). (**D**) A fragment shown in (**C**) (outlined by red rectangle) at a higher magnification. (**E**) Medial part of CLTNZ containing type III and IVa cells localized perpendicular to intertectal ventricle (indicated by green arrows). (**F**) Patterns of type III cell proliferation (daughter cell shown in red) in the apical layers of CLTNZ. Transmission electron microscopy. Scale: (**A**,**C**,**E**) 10 μm and (**B**,**D**,**F**) 2 μm.

**Figure 10 ijms-26-00644-f010:**
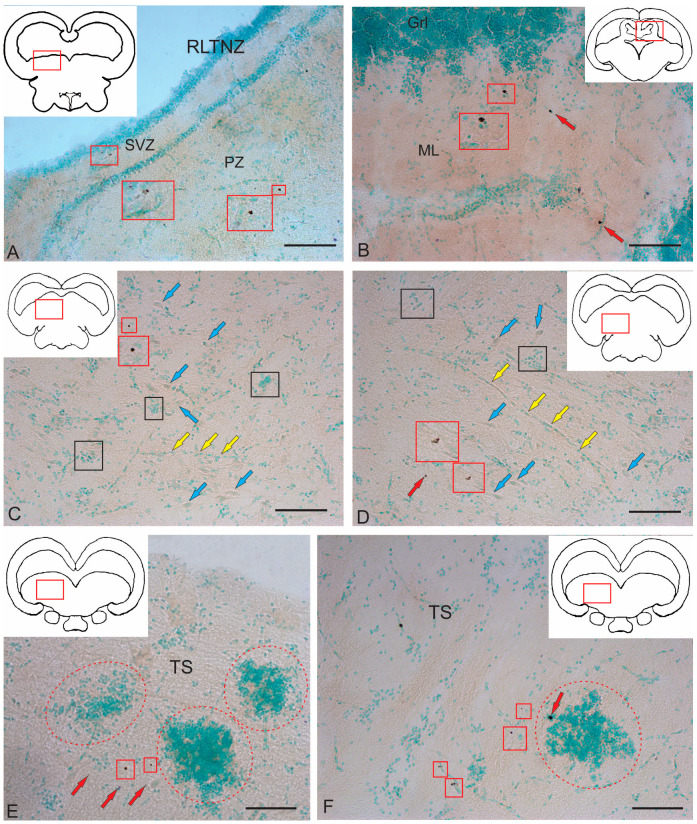
Immunohistochemical labeling of brain lipid-binding protein (BLBP) in the mesencephalic tegmentum of intact juvenile chum salmon, *Oncorhynchus keta*. (**A**) BLBP distribution in the rostral part of tegmentum of intact juveniles; in the periventricular region of RLTNZ, clusters of BLBP+ cells are in the red rectangle; SVZ—subventricular zone; PZ—parenchymatous zone. (**B**) In the *valvula cerebelli*, BLBP expression was detected in small type 1 cells (red arrows) located in the molecular layer (ML); enlarged fragment is located in the red rectangle. (**C**) At the rostro-medial level, projections of the nucleus of the III nerve, BLBP-negative interneurons of the III nerve (blue arrows), clusters of small cells (in black rectangles), and cells migrating along vessels (yellow arrows) are shown; BLBP+ cells are in the red rectangle. (**D**) The caudo-lateral level of the projection of the nucleus III nerve; designations are as in (**C**). (**E**) In the region of *torus semicircularis* (TS), single BLBP+ cells (red arrows) and large and dense BLBP-immunonegative clusters (in red dotted ovals) are shown. (**F**) In the ventral part of TS, single BLBP+ cells (in red rectangles) are shown. Scale: (**A**,**C**,**E**) 10 µm and (**B**,**D**,**F**) 2 µm.

**Figure 11 ijms-26-00644-f011:**
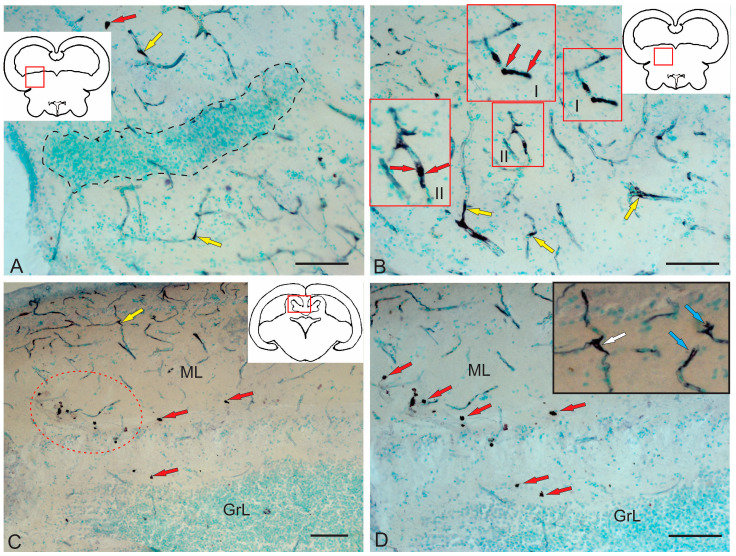
Immunohistochemical labeling of BLBP in the mesencephalic tegmentum of juvenile chum salmon, *Oncorhynchus keta*, with acute injury of *medulla oblongata*. (**A**) In the rostro-lateral part of the tegmentum, a cluster of small undifferentiated BLBP cells (outlined by dotted line), elongated BLBP+ cells (yellow arrows) along small vessels, and single parenchymatous BLBP+ cells (red arrow) are shown. (**B**) In the medial tegmentum, the migration of BLBP+ oval cells (red arrows) along vessels (shown in insets I and II in red rectangles) and elongated BLBP+ cells (yellow arrows) are shown. (**C**) In the *valvula cerebelli*, BLBP+ Bergmann glia fibers (yellow arrow) in the dorsal part of molecular layer (ML) are shown; the dotted red oval outlines an aggregation of BLBP+ oval cells; single BLBP+ cells are indicated by red arrows. (**D**) At a higher magnification, terminal branches of BLBP+ Bergmann glia (blue arrows) and BLBP+ cells (white arrow) migrating along Bergmann glia (black inset) are shown. Scale: (**A**,**C**) 10 μm and (**B**,**D**) 2 μm.

**Figure 12 ijms-26-00644-f012:**
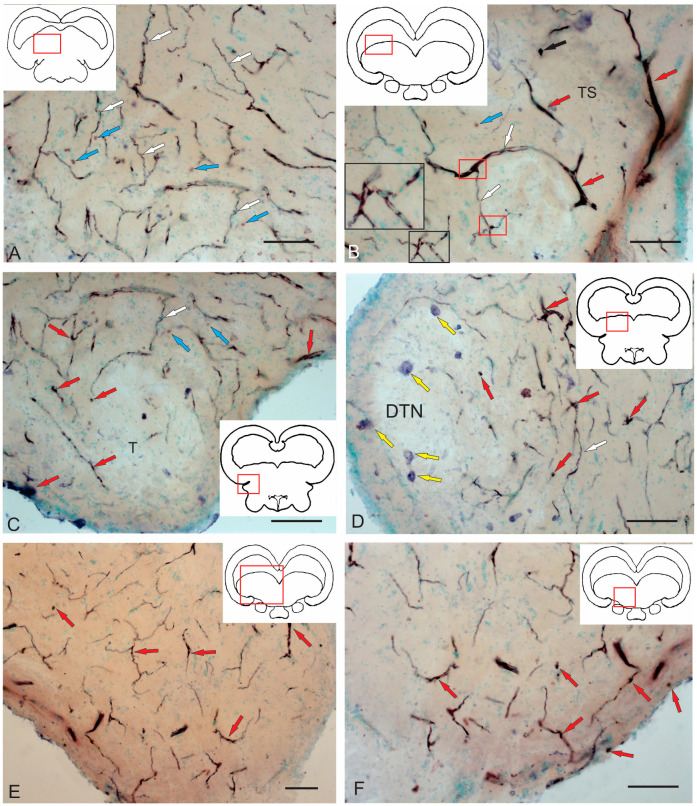
Immunohistochemical labeling of BLBP in radial glia (RG) of the tegmentum in juvenile chum salmon, *Oncorhynchus keta*, with acute injury of *medulla oblongata*. (**A**) BLBP immunolabeling in RG fibers (white arrows) at the NFLM level; blue arrows indicate BLBP+ cells in the tegmentum parenchyma. (**B**) In the area of *torus semicircularis* (TS), thick guides are indicated by red arrows; BLBP+ RG fibers are indicated by white arrows; intensely labeled BLBP+ cells (black arrow), parenchymal weakly labeled BLBP+ cells are indicated by blue arrows; multicellular complexes with BLBP+ and BLBP− cells are shown in the black rectangle; and terminal branches of BLBP+ RG fibers are shown in red rectangles. (**C**) In the area of the lateral torus (LT), migrating BLBP+ cells are indicated by red arrows; other designations are as in (**A**). (**D**) In the dorsal tegmentum, moderate BLBP expression in large DTN neurons are indicated by yellow arrows; migrating and parenchymal BLBP+ cells are indicated by red arrows; and BLBP+ RG fibers are indicated by the white arrow. (**E**) In the caudal tegmentum, migrating and parenchymatous BLBP+ cells are indicated by red arrows. (**F**) In the ventro-lateral tegmentum, the designations are as in (**E**). Scale: (**A**–**F**) 100 μm.

**Figure 13 ijms-26-00644-f013:**
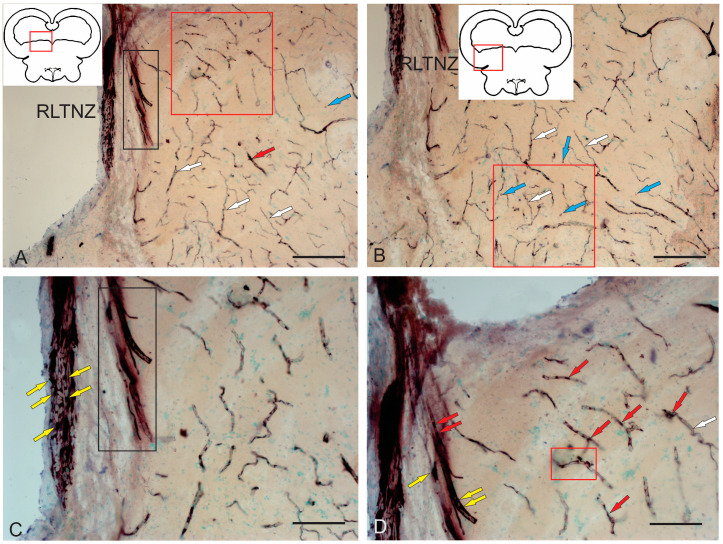
Immunohistochemical labeling of BLBP in the neurogenic zones of the tegmentum in juvenile chum salmon, *Oncorhynchus keta*, with acute injury of *medulla oblongata*. (**A**) In the RLTNZ, BLBP+ RG fibers are shown in the red rectangle; single immunolabeled RG fibers are indicated by white arrows; BLBP+ cells in the parenchyma are indicated by blue arrows; pattern of BLBP+ cell migration along RG is indicated by the red arrow; the black rectangle outlines proximal fragments of BLBP+ RG emerging from RLTNZ. (**B**) In the lateral tegmentum, in the red square, a fragment containing BLBP+ cells is indicated by blue arrows and fibers are indicated by white arrows. (**C**) An enlarged dorsal fragment is shown in (**A**); yellow arrows indicate BLBP+ in NSPCs in the RLTNZ. (**D**) An enlarged dorsomedial fragment in (**A**) containing BLBP+ RG (yellow arrows) and BLBP+ cells migrating along RG (red arrows); the red rectangle outlines the terminal branches of BLBP+ fibers; the white arrow indicates a thin BLBP+ branch of RG fiber. Scale: (**A**,**B**) 200 μm and (**C**,**D**) 100 μm.

**Figure 14 ijms-26-00644-f014:**
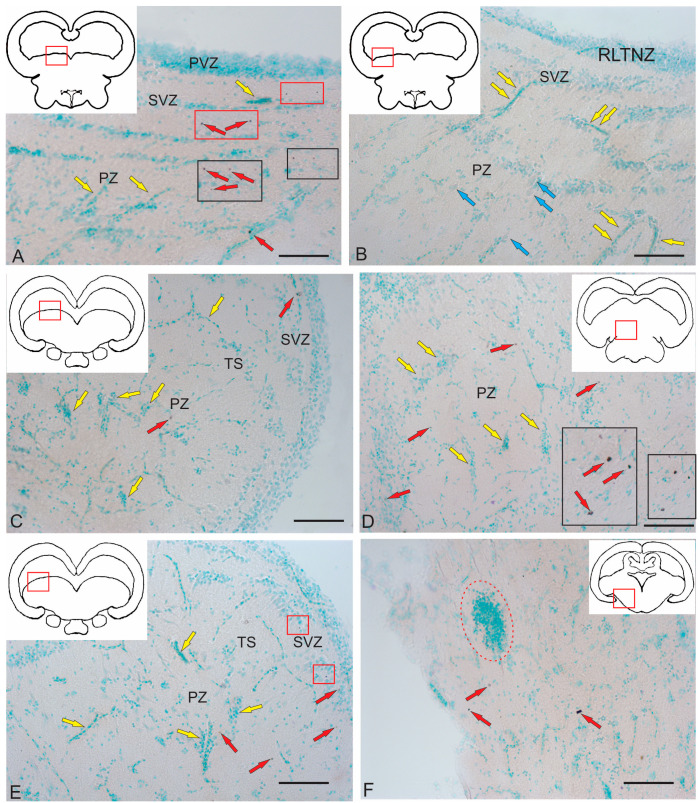
Immunohistochemical labeling of AroB in the tegmentum of intact juvenile chum salmon, *Oncorhynchus keta*. (**A**) In the SVZ in the dorsomedial region (red inset), dorsal parenchymal regions of PZ are shown in the black inset, AroB+ cells are indicated by red arrows, and migration patterns of immunonegative cells are indicated by yellow arrows. (**B**) In the dorsolateral region of tegmentum, immunonegative neurons are indicated by blue arrows and the migration of immunonegative cells is indicated by yellow arrows. (**C**) In the rostral part of TS, the designations are as in (**A**). (**D**) in NIII, AroB expression in cells of the medio-basal region is shown in the black rectangle; other designations are as in (**A**). (**E**) In the caudal part of TS, AroB expression in granule-like cells of SVZ is shown in the red rectangles; other designations are as in (**A**). (**F**) In the ventrolateral part of isthmus, AroB+ cells are indicated by red arrows and single immunonegative neurogenic niches are shown in the red dotted oval. Scale: (**A**–**F**) 100 μm.

**Figure 15 ijms-26-00644-f015:**
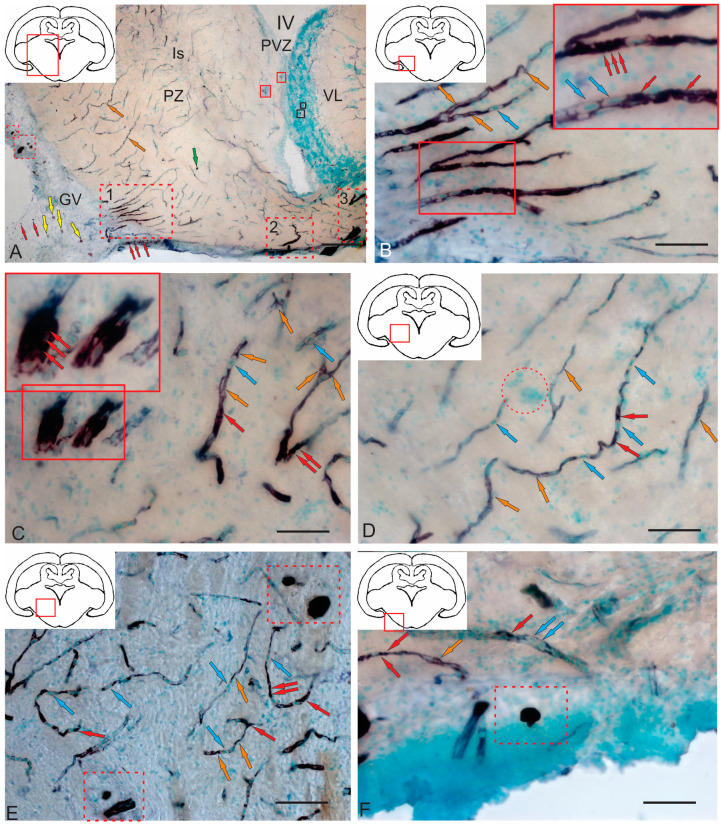
Immunohistochemical labeling of AroB in the tegmentum of juvenile chum salmon, *Oncorhynchus keta*, at 3 days after traumatic injury to *medulla oblongata*. (**A**) In the rostral isthmus (Is), AroB in populations of migrating cells from the basal and basolateral areas of the brain are outlined by red dotted rectangles 1, 2, and 3; in the trigeminal ganglion (GV) and pial membrane large, moderately AroB+ labeled cells are indicated by yellow arrows; small, intensely AroB+ labeled cells are indicated by red arrows; mixed clusters of intensely AroB+ labeled large and small cells are shown in the red dotted square; in the basal and basolateral part of isthmus, AroB+ fibers RG are indicated by orange arrows; in the PZ of isthmus, intensely AroB+ labeled medium-sized cells are indicated by green arrows; in PVZ, small intensely AroB+ cells are shown in the red square; AroB+ cells in the granular layer of cerebellar valve VL are shown in the black square. (**B**) Enlarged fragment is shown in the red dashed rectangle 1; AroB+ cells (red arrows); AroB-negative cells (blue arrows); AroB+ RG fibers (orange arrows). (**C**) In the ventro-lateral regions of the tegmentum, dense patterns of migration of intensely AroB+ labeled cells are shown in the red inset with AroB+ RG fibers; RG end feet converging on immunonegative cells are indicated by orange arrows; other designations are as in (**B**). (**D**) In the ventro-medial part of tegmentum, extended fragments of RG fibers with AroB+ and AroB− cells migrating along them are shown; clusters of AroB− cells are shown in the red dotted circle; other designations are as in (**B**). (**E**) In the baso-medial part of tegmentum, dense clusters of small intensely labeled AroB+ cells are shown in red dotted rectangle; other designations are as in (**B**). F—Secondary neurogenic niches in the outer marginal zone of tegmentum, on the border with the pial membrane are shown in in red dotted rectangle; other designations are as in (**B**). Scale: (**A**) 200 μm; (**B**–**E**) 100 μm; and (**F**) 50 μm.

**Figure 16 ijms-26-00644-f016:**
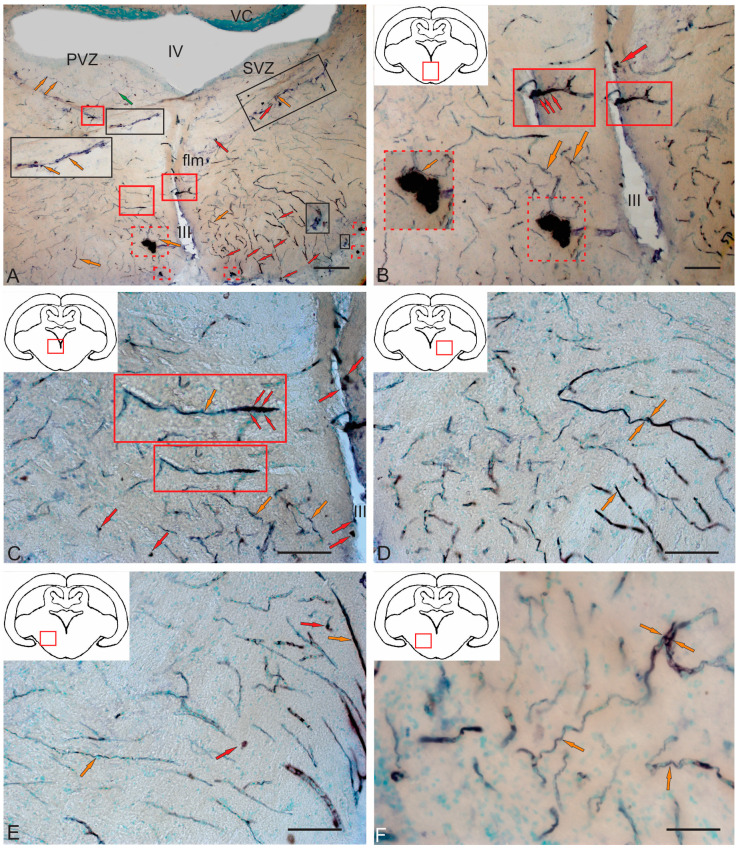
Immunohistochemical labeling of AroB in the caudal isthmus of juvenile chum salmon, *Oncorhynchus keta*, at 3 days after traumatic injury to *medulla oblongata*. (**A**) General view of isthmus with immunolabeling of AroB+ cells in secondary post-traumatic niches (red dotted rectangle); a linear cluster of AroB+ cells along the lateral wall of hypothalamic ventricle (in red rectangle); patterns of AroB+ cell migration along RG (in black rectangles); AroB+ cells (red arrows); AroB+ RG fibers (orange arrows); AroB-negative neurons of DTN (green arrow). (**B**) Enlarged fragment with secondary neurogenic niches containing AroB+ cells; designations as in (**A**). (**C**) Enlarged fragment along *infundibulum* with AroB+ cells in the pereinfundibular area (red arrows) and horizontal clusters of AroB+ cells (in red rectangles). (**D**) On the ipsilateral side of AroB+ RG (orange arrows), end apparatuses and fragments of fibers in the ventrolateral isthmus are shown. (**E**) On the contralateral side, single AroB+ cells are indicated by red arrows. (**F**)—In the ventro-medial part of the tegmentum (ipsilateral side), patterns of AroB+ RG and their interactions with migrating cells are indicated by orange arrows. Scale: (**A**) 200 μm; (**B**–**E**) 100 μm; and (**F**) 50 μm.

**Figure 17 ijms-26-00644-f017:**
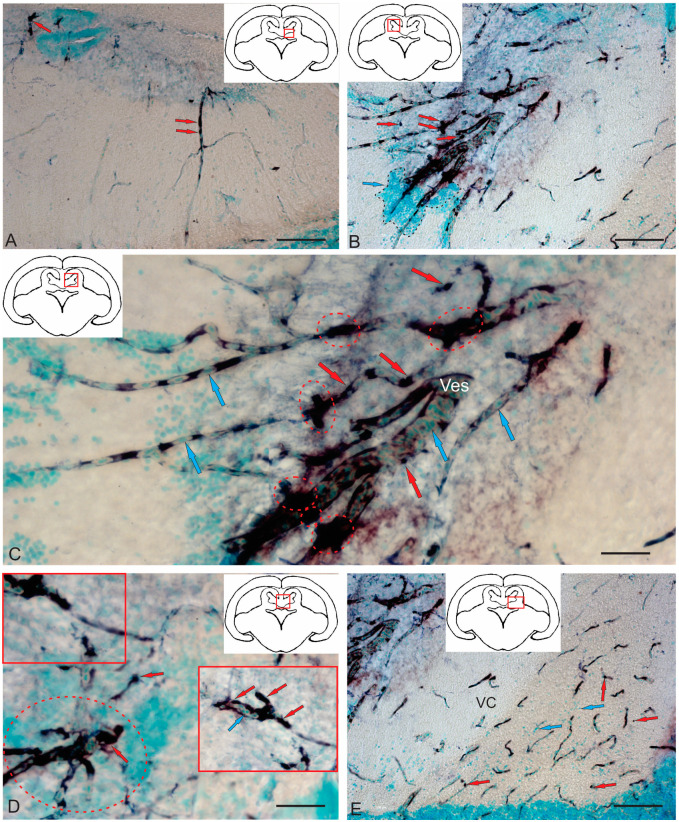
Immunohistochemical labeling of AroB in *valvula cerebelli* of juvenile chum salmon, *Oncorhynchus keta*, at 3 days after traumatic injury to the *medulla oblongata*. (**A**) Caudal projection of *valvula cerebelli*; migrating AroB+ cells along radial fibers of Bergmann glia (red arrows). (**B**) Mass patterns of AroB+ cell migration in the rostral part of *valvula cerebelli* from the molecular layer (ML); large clusters (blue arrow) of small immunonegative cells (outlined by black dotted line). (**C**) Central vascular complex (Ves) with superficial clusters of AroB+ cells (in red dotted ovals); immunonegative vascular endothelial cells (blue cells); AroB+ cells (red arrows). (**D**) Vascular structures with clusters of migrating AroB+ cells localized along them (in red rectangles); other designations are as in (**C**). (**E**) Patterns of AroB+ cell migration and fragments of AroB+ Bergmann glia in the peripheral zones of *valvula cerebelli* (VC) in the ML; designations are as in (**C**). Scale: (**A**,**B**,**E**) 100 μm and (**C**,**D**) 50 μm.

**Figure 18 ijms-26-00644-f018:**
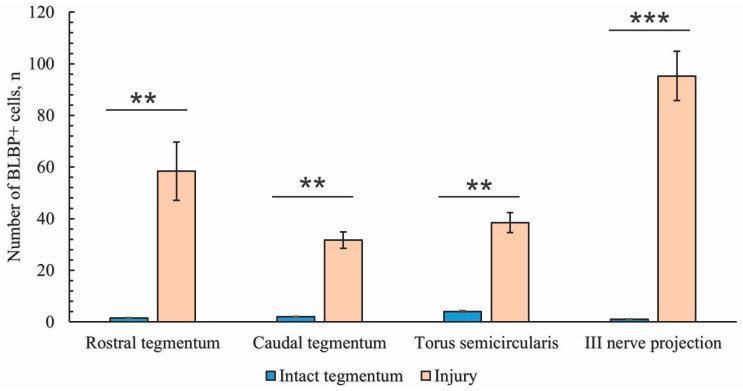
Quantitative proportions of BLBP+ cells in different tegmental areas of juvenile chum salmon, *Oncorhynchus keta*, at control and 3 days after traumatic injury to the *medulla oblongata* (M ± SD), where M is the mean and SD is the standard deviation (*n* = 5 in each group; ** *p* ≤ 0.01 and *** *p* ≤ 0.001—significant differences vs. control groups). Student–Newman–Keuls test.

**Figure 19 ijms-26-00644-f019:**
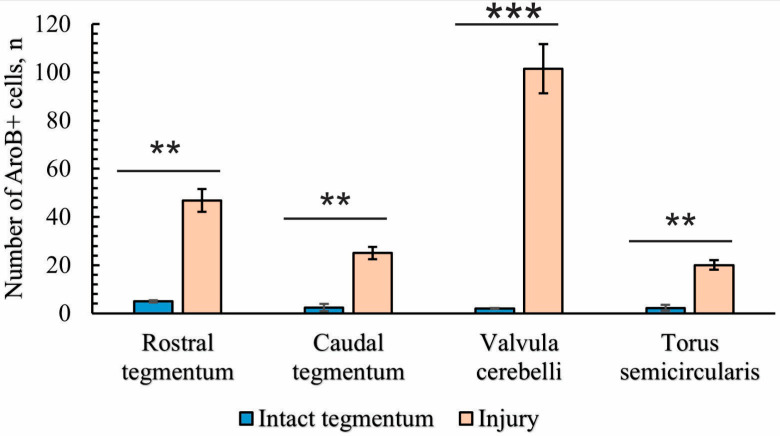
Quantitative proportions of Aro B+ cells in different tegmental areas of juvenile chum salmon, *Oncorhynchus keta*, at control and 3 days after traumatic injury to the *medulla oblongata* (M ± SD), where M is the mean and SD is the standard deviation (*n* = 5 in each group; ** *p* ≤ 0.01 and *** *p* ≤ 0.001—significant differences vs. control groups). Student–Newman–Keuls test.

**Figure 20 ijms-26-00644-f020:**
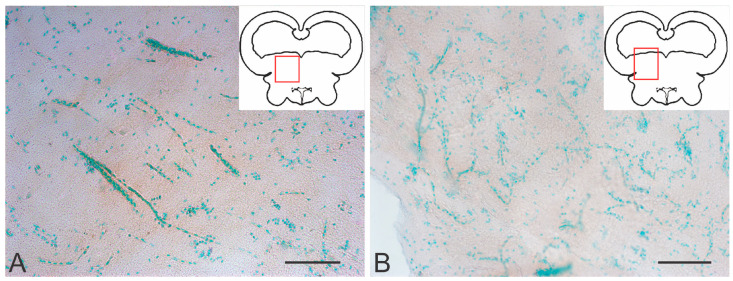
Negative controls (representative sections) in the tegmentum of juvenile chum salmon, *Oncorhynchus keta*. (**A**) IHC labeling of BLBP in the rostro-medial tegmentum. (**B**) IHC labeing of aromatase B in the rostro-lateral tegmentum. The red inset in the pictogram shows the respective zone in the micrograph. Scale bar: 100 μm.

**Table 1 ijms-26-00644-t001:** Ultrastructural characteristics of tegmentum nuclei in juvenile chum salmon, *O. keta*.

Type of Cells	Tegmental Neurons		
	Large InI	Medial InII	Small InIII
Sample size	*n* = 24	*n* = 22	*n* = 21
**Nucleus nervus oculomotoris NIII**			
Long axis of cells soma (µm)	21.15 ± 4.43	17.29 ± 2.63	9.47 ± 0.39
Short axis of cells soma (µm)	13.97 ± 0.46	8.4 ± 1.01	5.33 ± 0.27
**Nucleus**	1	1	1
Contour	ovoid	ovoid; irregular ± invaginations	ovoid; elongated
Long axis (µm)	10.25 ± 0.52	8.69 ±0.93	5.46 ± 0.63
Short axis (µm)	6.71 ± 0.83	5.45 ± 0.63	3.61 ± 0.53
Chromatin	reticulated euchromatin	reticulated; clumped heterochromatin	clumped euchromatin, heterochromatin
Color	medium-light	medium	medium
Nucleoli	1 or 2	1 or 2, rarely visible	1
**Cytoplasm**			
Percentage/color	abundant/light	abundant/light	medium/light
Mitochondria	many	few	few
Vacuoles	many; small	few	few; small
Lipid droplets	yes	yes	yes
Dense bodies	yes	yes	no
Cell contacts	yes	no	no
**Nucleus of fasciculus longitudinalis medialis (NFLM)**			
Sample size	*n* = 26	*n* = 20	*n* = 23
Long axis of cells soma (µm)	15.72 ± 0.83	11.41 ± 1.61	9.08 ± 0.29
Short axis of cells soma (µm)	8.96 ± 1.21	6.02 ± 0.91	6.13 ± 0.22
**Nucleus**	1–2	1	1
Shape	ovoid; irregular ± invaginations	ovoid; irregular ± invaginations	ovoid; irregular
Long axis (µm)	9.45 ±1.47	6.89 ± 0.96	7.77 ± 0.49
Short axis (µm)	5.49 ±1.44	3.55 ± 0.52	5.69 ± 0.32
Chromatin	some clumped	more clumped	heterochromatin
Color	light	light	light
Nucleoli	1 or 2	1 or 2	1 or more
**Cytoplasm**			
Percentage/color	70–80%, light gray or gray	50–60%, denser, gray	20–30%, light, grainy
Mitochondria	many	few	few
Vacuoles	few	few	few; small
Lipid droplets	many	fewer but larger	single
Dense bodies	yes	yes	single
Cell contacts	large synaptic terminals	with InII	-
**Dorsal tegmental nucleus DTN**			
Sample size	*n* = 21	*n* = 25	*n* = 20
Long axis of cells soma (µm)	47.95 ± 5.3	11.87 ± 1.17	5.85 ± 0.25
Short axis of cells soma (µm)	22.24 ± 2.59	5.35 ± 0.87	3.33 ± 0.13
**Nucleus**	1 or no	1	1
Contour	ovoid	ovoid; irregular ± invaginations	-
Long axis (µm)	12.47 ± 2.45	7.62 ± 1.04	4.66 ± 0.34
Short axis (µm)	7.26 ± 0.81	4.18 ± 0.75	2.92 ± 0.29
Chromatin	evenly distributed; non-clumped	heterochromatin	heterochromatin
Color	light or dark	light or medium	medium, coarse grain
Nucleoli	1	1	few
**Cytoplasm**			
Percentage/color	70–80%, dense gray or gray	60–70%, dense, coarse-grained	20–30%, gray, fine-grained
Mitochondria	many	intermediate	few
Vacuoles	few	few	few
Lipid droplets	many	few	no
Dense bodies	yes	yes	yes
Cell contacts	with astrocytes	-	-

**Table 2 ijms-26-00644-t002:** Ultrastructural characteristics of cells of the periventricular zone in the tegmentum of juvenile chum salmon, *O. keta*.

Type of Cells	Ependyma	III	IVa	Neuroepithelial Cells
Sample size	*n* = 32	*n* = 51	*n* = 59	*n* = 7
Long axis of cells soma (µm)	1.73 ± 0.42	7.22 ± 0.83	6.72 ± 0.04	11.36 ± 0.39
Short axis of cells soma (µm)	0.99 ± 0.17	3.93 ± 0.69	5.66 ± 0.09	4.23 ± 0.03
**Nucleus**				
Shape	elliptical; elongated ± few invaginations	elongated; irregular ± invaginations	ovoid, irregular ± invaginations	ovoid, elongated ± invaginations
Long axis (µm)	0.78 ± 0.06	6.62 ± 0.47	5.85 ± 0.19	5.28 ± 0.38
Short axis (µm)	0.52 ± 0.07	3.35 ± 0.29	4.79 ± 0.21	2.52 ± 0.25
Chromatin	evenly distributed; heterochromatin	evenly distributed; non-clumped	reticulated; clumped hetero	reticulated; clumped hetero.
Color	light	medium	dark	medium
Nucleoli	1 or 2	1 or 2	1 or 2, rarely visible	1 or 2
**Cytoplasm**				
Percentage/color	abundant/light	scanty/light	scanty/medium	abundant/medium
Mitochondria	many	few	few	few
Microvilli	few to many	no	no	few
Vacuoles	Many, large	no	few	few
Lipid droplets	yes	no or 1	no	no
Dense bodies	yes	yes	no	yes
**Localization**	RLTNZ, TMZ	RLTNZ, TSNZ, CLTNZ	RLTNZ, TSNZ, CLTNZ	RLTNZ
**Contacts**	DEL, capillaries	III, IVa	III, IVa	DEL, ependyma

**Table 3 ijms-26-00644-t003:** Morphometric and densitometric characteristics of BLBP labeled cells (M ± SD) in the intact and injured *medulla oblongata* of juvenile chum salmon, *O. keta*.

Brain Areas	Type of Cells	Cell Size, μm *	Optical Density **, UOD
**Intact fish**			
Rostral tegmentum, SVZ	Elongated	4.02 ± 0.54/2.28 ± 0.4 5.75 ± 0.6/4.4 ± 0.4	68 ± 24.04 87 ± 8.8
NIII	Small Elongated	4.44 ± 0.55/3.32 ± 0.16 8.24 ± 0.7/6.61 ± 0.6	54.5 ± 0.71 82 ± 8.6
*Torus semicircularis*	Small Elongated	2.76 ± 0.57/2.37 ± 0.64 8.36 + 0.8/5.89 ± 0.6	56.75 ± 30.78 96.67 ± 34.85
*Valvula cerebelli*, ML	Oval Elongated	6 ± 0.6/3.84 ± 0.4 8.59 ± 0.93/6.65 ± 0.35	95.5 ± 23.33 86 ± 8.6
**Traumatic injury of *medulla oblongata***			
Rostral tegmentum	Small Migrating Oval Elongated	4.65 ± 0.55/2.83 ± 0.81 7.04 ± 1.29/2.95 ± 0.89 6.72 ± 6.72/3.75 ± 3.75 6.38 ± 6.38/4.62 ± 4.62	135.8 ± 13.75 142.4 ± 15.44 132.4 ± 20.31 149 ± 8.25
Caudal tegmentum	Small Migrating Oval	4.76 ± 0.82/3.28 ± 1.11 8.01 ± 1.65/3.28 ± 0.59 6.932 ± 1.03/3.52 ± 0.7	132.8 ± 19.4 143.43 ± 10.21 132.8 ± 13.03
NIII	Rounded Small Migrating Oval Elongated	5.61 ± 0.5/5.53 ± 0.4 5 ± 0.39/2.98 ± 0.49 8.61 ± 1.11/3.24 ± 0.99 6.26 ± 1.04/3.89 ± 0.99 6.54 ± 0.44/4.98 ± 0.5	119 ± 45.12 135.2 ± 26.7 153.3 ± 14.22 119.4 ± 42.86 133.8 ± 36.77
*Torus semicircularis*	Small Migrating	4.23 ± 0.67/2.31 ± 0.86 7.17 ± 2.05/2.97 ± 0.27	151.8 ± 16.38 160 ± 8.09

* The values before and after slash (/) are for large and small diameters of the cell body. ** Optical density (OD) in cells was classified according to the following scale: high (160–130 UOD), moderate (130–100 UOD), and weak (less than 100 UOD).

**Table 4 ijms-26-00644-t004:** Morphometric and densitometric characteristics of AroB labeled cells (M ± SD) in the intact and injured *medulla oblongata* of juvenile chum salmon, *O. keta*.

Brain Areas	Type of Cells	Cell Size, μm *	Optical Density **, UOD
**Intact fish**			
Rostral tegmentum, SVZ Medio-basal tegmentum	Small, Undifferentiated	3.79 ± 0.84/2.82 ± 0.4	63 ± 16.27
NIII	Oval Rounded	8.24 ± 1.41/4.88 ± 0.23 5.79 ± 0.6/4.57 ± 0.5	106.75 ± 9.74 48.33 ± 10.02
*Torus semicircularis*, SVZ PZ	Small, Undifferentiated	4.38 ± 1.47/3.22 ± 1.03	56.5 ± 6.36
**Traumatic injury of *medulla oblongata***			
Rostral isthmus	Large Small Migrating Oval Elongated	19.24 ± 3.25/13.79 ± 3.17 4.11 ± 0.47/2.35 ± 0.63 7.64 ± 1.67/2.84 ± 1.25 6.11 ± 0.68/3.5 ± 0.5 5.94 ± 0.6/4.8 ± 0.4	65 ± 30.95 150.75 ± 9.36 95.2 ± 44.91 152 ± 11.8 159.5 ± 2.38
Caudal isthmus	Small Migrating Oval Elongated	4.7 ± 1.11/2.56 ± 0.63 8.93 ± 0.87/3.15 ± 1.05 6.79 ± 1.2/3.26 ± 0.6 7.65 ± 2.29/5.81 ± 1	134 ± 40.48 124.82 ± 26.49 140.8 ± 18.9 128.14 ± 16.19
*Valvula cerebelli*, Granular layer	Small Migrating Oval Elongated	4.33 ± 0.85/2.21 ± 0.11 7.42 ± 1.75/2.67 ± 0.54 8.26 ± 2.07/4.94 ± 1.25 0.32 ± 6.52/0.01 ± 5.45	149.4 ± 15.06 150.5 ± 9.43 142.29 ± 19.32 148 ± 15.62

* The values before and after slash (/) are for large and small diameters of the cell body. ** Optical density (OD) in cells was classified according to the following scale: high (160–130 UOD), moderate (130–100 UOD), and weak (less than 100 UOD).

**Table 5 ijms-26-00644-t005:** Characteristics of primary antibodies used in immunohistochemical studies.

No.	Antibodies	Manufacturer	Dilution	Catalog Number	Marker
1	BLBP	Abcam, Cambridge, UK	1:300	ab32423	BLBP
2	AroB (cyp19a1b)	Abcam, Cambridge, UK	1:300	ab106168	AroB (cyp19a1b)

## Data Availability

Data are contained within the article and Appendix A.

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
