# Peer review of "Ultrastructural Study and Immunohistochemical Characteristics of Mesencephalic Tegmentum in Juvenile Chum Salmon (Oncorhynchus keta) Brain After Acute Traumatic Injury"

_ijms, 2025, doi:10.3390/ijms26020644_

Round 1
Reviewer 1 Report
Comments and Suggestions for Authors
In the paper entitled " Ultrastructural study and immunohistochemical characteristics of mesencephalic tegmentum in juvenile chum salmon (Oncorhynchus keta) brain after acute traumatic injury" the authors provide a detailed ultrastructural description of the cellular characteristics forming the tegmental nuclei in juvenile chum salmon. They also present an immunohistochemical analysis of BLBP and AroB expression levels in the tegmentum of intact specimens and during the post-traumatic period following acute injury to the medulla oblongata. However, the study raises several concerns:
1. Figure 1: A higher magnification is required since the nuclei described are not clearly visible. Additionally, defining all cells as aNSPCs or neurons without the use of a specific marker is speculative.
2. Table 1: The columns "Long axis of cells soma (μm)" and "Short axis of cells soma (μm)" are repeated and should be corrected. Furthermore, the titles of Tables 1 and 2 should be revised to change "Morphological" to "Ultrastructural".
3. Electron Microscopy Figures: The colors used to indicate different cell types or areas of interest overly obscure the structures described, making them difficult for readers to identify. The authors should either provide the original unmodified images alongside the annotated ones or increase the transparency of the overlays.
4. Figure 3: A higher magnification is needed for the synaptic terminals, oligodendrocytes, and apoptotic cells (panel A). In panel B, the black arrows indicating the nucleolus are not visible, and the oligodendrocytes require greater magnification.
5. Line 465: The term "inset" should be corrected.
6. Figure 8 (panel E): The white arrows and the label "Nu" in the cell nucleus are not visible and should be revised.
More significant corrections, however, relate to the bright-field immunohistochemistry section. The images provided often have low resolution, making it difficult to clearly identify immunopositivity. In many cases, the staining appears as nonspecific precipitate or debris on the histological samples (e.g., Figures 11, 12, 13, 14, and 16). Furthermore, the observed immunopositivity seems more indicative of nonspecific labeling than a clear identification of the investigated markers.
This hypothesis is supported by the sample processing method described in the Materials and Methods section. The authors mention the use of two mouse monoclonal antibodies, anti-BLBP and anti-AroB. However, according to the supplier's website, these antibodies are rabbit polyclonal antibodies. The authors should clarify the RRIDs of the reagents used in the immunohistochemistry experiments.
Moreover, the two antibodies used are validated for Zebrafish but may not be specific for the species studied, as the results suggest. Additionally, while the anti-BLBP antibody is validated for paraffin-embedded immunohistochemistry, it does not appear to work on cryopreserved samples (the experimental condition employed in this study).
Lastly, a statistical analysis of the collected results should be considered.
In conclusion, while the study is well-structured and innovative, numerous issues must be addressed before it can be considered for publication
Author Response
In the paper entitled " Ultrastructural study and immunohistochemical characteristics of mesencephalic tegmentum in juvenile chum salmon (Oncorhynchus keta) brain after acute traumatic injury" the authors provide a detailed ultrastructural description of the cellular characteristics forming the tegmental nuclei in juvenile chum salmon. They also present an immunohistochemical analysis of BLBP and AroB expression levels in the tegmentum of intact specimens and during the post-traumatic period following acute injury to the medulla oblongata. However, the study raises several concerns:
Thank You for reviewing our work and for your valuable comments. We agree with most of the comments of the distinguished reviewer and thank you for the critical analysis of our article. The corrected version of the paper took into account the comments and the corresponding corrections were made to the edited version of the article. The corrected fragments of the work are highlighted in red font.
- Figure 1: A higher magnification is required since the nuclei described are not clearly visible. Additionally, defining all cells as aNSPCs or neurons without the use of a specific marker is speculative.
We thank You for this comment. However, Figure 1 shows photographs of semi-thin sections of the tegmentum in order to demonstrate the general organization of this area of the brain, without detailing the nuclear structure. The photographs in Fig. 1 show the main areas of sections of the brain of juvenile chum salmon at the level of the tegmentum. The purpose of this demonstration was not to present the studied nuclei using TEM, therefore, the semifine slices of the brain only partially show the tegmentum nuclei studied using TEM. Clarifications have been added to the caption (instead of aNSCP, it has been changed to " probably aNSPCs").
- Table 1:The columns "Long axis of cells soma (μm)" and "Short axis of cells soma (μm)" are repeated and should be corrected. Furthermore, the titles of Tables 1 and 2 should be revised to change "Morphological" to "Ultrastructural".
Thank You for this comment, corrections have been made to the titles of Tables 1 and 2. Corrections have been made to Table 1, the "Long axis of the nucleus" and the "Short axis of the nucleus " have been changed, respectively.
- Electron Microscopy Figures:The colors used to indicate different cell types or areas of interest overly obscure the structures described, making them difficult for readers to identify. The authors should either provide the original unmodified images alongside the annotated ones or increase the transparency of the overlays.
Thank You for this comment. The original images of the TEM photographs for Figures 2-5, 7,9 (without applying a color mask) are attached to "Suplimentary files" to improve the perception of structures.
- Figure 3:A higher magnification is needed for the synaptic terminals, oligodendrocytes, and apoptotic cells (panel A). In panel B, the black arrows indicating the nucleolus are not visible, and the oligodendrocytes require greater magnification.
Thank You for this comment. In Fig. 3, the enlarged organization of the synaptic structure of the contralateral part of the tegmental nucleus is shown in panels C-F. The main objective of this illustration was to demonstrate the structure of an asymmetric synapse on the contralateral side of the NFLM. A general view of the ipsilateral side is shown in panel A. Enlarged images of the apoptotic cell and oligodendrocyte are shown in the edited version of Fig. 3 B.
- Line 465:The term "inset" should be corrected.
Thanks for this comment, corrections have been made.
- Figure 8 (panel E):The white arrows and the label "Nu" in the cell nucleus are not visible and should be revised.
Thank You for this comment, the relevant corrections have been made in Fig. 8E.
More significant corrections, however, relate to the bright-field immunohistochemistry section. The images provided often have low resolution, making it difficult to clearly identify immunopositivity. In many cases, the staining appears as nonspecific precipitate or debris on the histological samples (e.g., Figures 11, 12, 13, 14, and 16). Furthermore, the observed immunopositivity seems more indicative of nonspecific labeling than a clear identification of the investigated markers.
Thank You for this comment. We have presented the Figures 11, 12, 13, 14, 16, high-resolution IHC labeling in a light field.
This hypothesis is supported by the sample processing method described in the Materials and Methods section. The authors mention the use of two mouse monoclonal antibodies, anti-BLBP and anti-AroB. However, according to the supplier's website, these antibodies are rabbit polyclonal antibodies. The authors should clarify the RRIDs of the reagents used in the immunohistochemistry experiments.
Thank You for this comment. Unfortunately, a technical error was made during the preparation of the Materials and Methods section. Indeed, polyclonal rabbit antibodies with specificity for zebrafish were used in the work. Appropriate changes have been made to the Materials and Methods section. Previously, data with antibodies against aromatase B were presented in the article Pushchina EV, Bykova ME, Varaksin AA. Post-Traumatic Expressions of Aromatase B, Glutamine Synthetase, and Cystathionine-Beta-Synthase in the Cerebellum of Juvenile Chum Salmon, Oncorhynchus keta. Int J Mol Sci. 2024 Mar 14;25(6):3299. doi: 10.3390/ijms25063299. PMID: 38542274; PMCID: PMC10970380.. In this work, similar antibodies against aromatase B were used. Negative control was performed to control the specificity of the antibodies. In the absence of specific antibodies, there was no IHC reaction. The negative control data is shown in Figure 20.
Moreover, the two antibodies used are validated for Zebrafish but may not be specific for the species studied, as the results suggest. Additionally, while the anti-BLBP antibody is validated for paraffin-embedded immunohistochemistry, it does not appear to work on cryopreserved samples (the experimental condition employed in this study).
Thank You for this clarification. The specification for antibodies indicates that the IHC reaction can be carried out on paraffin samples, however, the use of antibodies on cryopreserved samples implies less additional processing associated with pouring into paraffin, which can lead to antigen masking and requires an additional procedure for "unmasking" the antigen. There are no such restrictions for cryostatic and vibratomic sections.
Lastly, a statistical analysis of the collected results should be considered.
Thank You for this recommendation. The corrected version of the article includes data from the statistical analysis of Apo B and BL BP immunopositive cells in tegmental areas in intact animals and after traumatic injury (Figures 18, 19).
In conclusion, while the study is well-structured and innovative, numerous issues must be addressed before it can be considered for publication
Thank You for this offer. In accordance with the recommendations, significant changes have been made to the revised version of the article.

Reviewer 2 Report
Comments and Suggestions for Authors
Dear Authors,
The manuscript titled “Ultrastructural study and immunohistochemical characteristics of mesencephalic tegmentum in juvenile chum salmon (Oncorhynchus keta) brain after acute traumatic injury” is well constructed and written. The study focused on a very interesting and emerging field of research but some concerns are present.
On this regard, the exploration of neurogenic zones and adult neural stem progenitor cells in the context of acute traumatic injury contributes valuable insights to neurogenesis and brain plasticity research. The methodology is rigorous concerning the use of advanced techniques such as TEM and immunohistochemical labelling that ensures detailed ultrastructural and molecular insights. Moreover, the incorporation of morphometric and densitometric analyses provides quantitative support to the observations, enhancing the scientific rigor and effectively supports the textual content.
Major concerns: Although the study references zebrafish and other models, the extrapolation of findings to other species, including mammals, remains limited. Moreover, the potential translational implications of the findings for broader neuroregeneration research are not sufficiently discussed. Another limitation of the present research is the small sample size for morphometric and densitometric analyses that is not explicitly detailed, potentially limiting statistical power and generalizability. Furthermore, the research need functional assay (such as behavioral assessments) that could validate the impact of observed cellular changes. Finally, the study focuses on a single post-injury time point (3 days), which limits understanding of the dynamics of neurogenesis and glial responses over time. All this limitations should be discussed.
Minor concerns: Some classifications, such as the "embryonization phenomenon," require more thorough definitions and justification. Moreover, the relationship between observed cellular morphologies and functional roles could be elaborated further.
Author Response
Dear Authors,
The manuscript titled “Ultrastructural study and immunohistochemical characteristics of mesencephalic tegmentum in juvenile chum salmon (Oncorhynchus keta) brain after acute traumatic injury” is well constructed and written. The study focused on a very interesting and emerging field of research but some concerns are present.
We thank the distinguished Reviewer for evaluating our work.
On this regard, the exploration of neurogenic zones and adult neural stem progenitor cells in the context of acute traumatic injury contributes valuable insights to neurogenesis and brain plasticity research. The methodology is rigorous concerning the use of advanced techniques such as TEM and immunohistochemical labelling that ensures detailed ultrastructural and molecular insights. Moreover, the incorporation of morphometric and densitometric analyses provides quantitative support to the observations, enhancing the scientific rigor and effectively supports the textual content.
Thank You for your appreciation and analysis of our work.
Major concerns: Although the study references zebrafish and other models, the extrapolation of findings to other species, including mammals, remains limited. Moreover, the potential translational implications of the findings for broader neuroregeneration research are not sufficiently discussed. Another limitation of the present research is the small sample size for morphometric and densitometric analyses that is not explicitly detailed, potentially limiting statistical power and generalizability. Furthermore, the research need functional assay (such as behavioral assessments) that could validate the impact of observed cellular changes. Finally, the study focuses on a single post-injury time point (3 days), which limits understanding of the dynamics of neurogenesis and glial responses over time. All this limitations should be discussed.
Thanks for Your comments. We agree with the proposed limitations of this work and the need to discuss the potential applied implications of the results of this work for broader research in the field of neuroregeneration. To eliminate these problems, the sections Conclusion and Limitations and Perspectives have been added to the article, in which we tried to take into account the comments made. We agree that the samples of neurons and non-neuronal cells are small, but nevertheless, we added statistical analysis data for IHC experiments on aromatase B and BLBP labeling in conditions of intact and post-traumatic injury.
We thank You for the recommendation to provide data on functional analysis. In accordance with this recommendation, we have added 3 video files showing intact animals (supplementary file 1), animals immediately after injury to the medulla oblongata (supplementary file 2) and animals 3 days after injury (supplementary file 3).
In the Conclusion section, we summarized data on ultrastructural analysis, as well as IHC analysis on juvenile chum salmon on various post-traumatic periods, including acute trauma, long-term healing trauma, and repeated trauma (data with aromatase B labeling on the cerebellum).
Minor concerns: Some classifications, such as the "embryonization phenomenon," require more thorough definitions and justification. Moreover, the relationship between observed cellular morphologies and functional roles could be elaborated further.
Thank You for this comment. We have added a definition and justification to the "phenomenon of embryonization," according to the commentary. In the Discussion section, "fetalization—the retention of features of the embryonic structure—in the brain of 2-year-old chum salmon at the ultrastructural level" was defined (pp. 1208-1209). The phenomenon of embryalization is associated with the preservation of features of the embryonic structure in adult animals, in our case in two-year-old fish, references have been added to the literature describing this phenomenon in salmonids.
Special thanks for the desire to examine in more detail the relationship between the observed morphological features of cells and their functional roles. In the Conclusion section, we presented video monitoring data for an intact animal, immediately after injury to the medulla oblongata and 3 days after injury to the medulla oblongata.

Round 2
Reviewer 1 Report
Comments and Suggestions for Authors
The authors have made all the necessary revisions to the manuscript, incorporating all the requested points and significantly improving the quality of the proposed work.
Reviewer 2 Report
Comments and Suggestions for Authors
Dear Authors,
the paper has been deeply improved and should be considered for publication.